# Demand Response Analysis Framework (DRAF): An Open-Source Multi-Objective Decision Support Tool for Decarbonizing Local Multi-Energy Systems

Markus Fleschutz [1,2], Markus Bohlayer [2], Marco Braun [2] and Michael D. Murphy [1,*]

1. Department of Process, Energy and Transport Engineering, Munster Technological University, T12 P928 Cork, Ireland; markus.fleschutz@h-ka.de
2. Institute of Refrigeration, Air-Conditioning, and Environmental Engineering, Karlsruhe University of Applied Sciences, Moltkestraße 30, 76133 Karlsruhe, Germany; markus.bohlayer@h-ka.de (M.B.); marco.braun@h-ka.de (M.B.)
* Correspondence: michaeld.murphy@mtu.ie

**Abstract:** A major barrier to investments in clean and future-proof energy technologies of local multi-energy systems (L-MESs) is the lack of knowledge about their impacts on profitability and carbon footprints due to their complex techno-economic interactions. To reduce this problem, decision support tools should integrate various forms of decarbonization measures. This paper proposes the Demand Response Analysis Framework (DRAF), a new open-source Python decision support tool that integrally optimizes the design and operation of energy technologies considering demand-side flexibility, electrification, and renewable energy sources. It quantifies decarbonization and cost reduction potential using multi-objective mixed-integer linear programming and provides decision-makers of L-MESs with optimal scenarios regarding costs, emissions, or Pareto efficiency. DRAF supports all steps of the energy system optimization process from time series analysis to interactive plotting and data export. It comes with several component templates that allow a quick start without limiting the modeling possibilities thanks to a generic model generator. Other key features are the access and preparation of time series, such as dynamic carbon emission factors or wholesale electricity prices; and the generation, handling, and parallel computing of scenarios. We demonstrate DRAF's capabilities through three case studies on (1) the DR of industrial production processes, (2) the design optimization of battery and photovoltaic systems, and (3) the design optimization and DR of distributed thermal energy resources.

**Keywords:** open-source; decarbonization; decision support; demand response; flexibility; electricity market; energy system modeling; multi-energy systems; optimization; smart energy technologies

## 1. Introduction

Following the Paris Agreement 2015 [1] and the Glasgow Climate Pact 2021 [2] there is an urgent need for decarbonization worldwide. With the European Green Deal approved in 2020, the European Union (EU) aims to achieve carbon neutrality by 2050 [3]. Energy system modeling is essential to driving the rapid adoption of smart energy services and clean innovative technologies needed to decarbonize energy systems [4]. Powerful energy system modeling frameworks have been presented as proprietary and open-source software. Most open-source frameworks, though, were developed with large-scale energy systems in mind, and only a few focus on decision support for local multi-energy systems (L-MESs). However, significant adoption of clean technologies and demand response (DR) programs are required in individual L-MESs, as the success of the energy transition depends on decentralization. According to the International Energy Agency, 70% of clean energy investments over the next decade will have to be made by private developers, consumers,

and financiers [5]. Industrial and commercial electricity consumers can contribute to the expansion of renewable energy sources (RES), e.g., through on-site photovoltaic systems. At the same time, they can help to integrate the RES through the smart use of flexible loads and energy storage through DR, which additionally offers cost-saving potential for companies. This demand-side flexibility is needed, as the flexibility demand of the electricity system is expected to quadruple by 2050, even though the availability of conventional flexibility sources will decrease due to the decommissioning of fossil power plants [6]. Therefore, in this paper, we focus on how cost and emission reductions that arise from applying DR to L-MESs, especially in the industrial and commercial sectors, can be quantified using the Demand Response Analysis Framework (DRAF).

Decarbonization of L-MESs are often considered by minimizing greenhouse gas emissions and costs within multi-objective modeling [7]. E.g., in [8], an efficient energy management model based on a multi-objective optimal power flow problem is proposed that considers flexibility of storage units of industrial networks. In [9], the contributions of energy storage systems, production buffer stocks, and smart transformers to a net zero energy factory were analyzed. Andiappan [10] reviewed mathematical optimization approaches for energy system synthesis and identified the concept of eco-industrial parks to be a future research direction.

### 1.1. Demand Response

Due to temporal variability, uncertainty, and location constraints, the integration of high shares of RES is demanding and requires high operational flexibility of the power system, which could lead to integration costs of RES [11]. In DR, final consumers provide demand-side flexibility to the electricity system by voluntarily changing their load profiles in reaction to price signals (price-based DR) or specific requests (incentive-based DR) [12]. It is widely acknowledged that DR is a key element of the transformation to a carbon-free energy system enabling cost-efficient integration of fluctuating RES [13]. Gils [14] identified a significant DR potential in Europe; the minimum aggregated hourly averages of load reduction and increase were 61 and 68 GW, respectively. For a general overview of demand response, the reader is referred to [15].

#### 1.1.1. DR in the Industrial and Commercial Sector

In recent years, several research projects investigating the DR potentials in the industrial and commercial sectors have been funded at the European and national level. Examples on European level are: demand response in industrial production (DRIP) [16], demand response integration technologies (DRIvE) [17], using the flexibility potential in energy intensive industries to facilitate further grid integration of variable renewable energy sources (IndustRE) [18]. Examples on national level are: SynErgie [19], Pilot Project DSM Bavaria [20], refrigerated warehouses store energy for smart energy grid (FlexLast) [21]. Commercial and industrial enterprises have large DR potentials [22], which can be divided into cross-sectional technologies [23] and energy-intensive production processes [24]. However, in order to exploit the full environmental and economic DR potential in the industrial and commercial sectors, aspects such as dynamic pricing [25], sector coupling, onsite generation, and onsite energy storage need to be considered in an integral analysis. Due to this complexity, there is still a lack of knowledge about existing flexibility, which is a major barrier to participation in DR programs [26]. To address this complexity, mathematical optimization, which has already been long proven in the analysis of large international power systems, can be applied to distributed energy systems. However, data preparation, model formulation, scenario definition, and result presentation require relevant experience and expertise to which the decision-makers in the companies often do not have access.

The aim of this study was, therefore, to develop and demonstrate the DRAF, which automates these process steps as far as possible to make the methodology of mathematical optimization accessible to a broader user group. DRAF is meant to supply insights and decision support for academics in applied sciences, consulting engineers, and decision-

makers in companies. Hence, it needs to be portable, easy-to-use, maintainable, editable, and extensible.

### 1.1.2. DR and Investments

The consideration of DR within optimal design planning generally results in higher capacities for assets and storage coupled to the electricity demand, since the associated lower average utilization rate is over-compensated for by the revenue or cost reductions from DR. Many analyses of DR potentials are limited to the potential of flexibility derived from existing assets. However, investment options that alter the existing flexibility, e.g., product storage extension for a flexible production process, or electrification measures, should also be considered when assessing the flexibility potential of L-MESs. E.g., Liu et al. [27] found synergistic effects when energy storage and DR of cooling, heating, and power were combined.

### 1.1.3. DR and Carbon Emissions

Electricity carbon emission factors (CEFs) can be categorized into grid-mix emission factors (XEFs) and marginal emission factors (MEFs). While XEFs are suitable for calculating carbon emission balances of energy consumption, MEFs are superior if the real carbon emission effects of a short-term demand change are to be approximated. Summerbell et al. [28] studied the cost and carbon reduction potentials of a cement plant through price-based DR using real-time prices (RTP). Although the carbon reduction potential was calculated from dynamic XEFs, they were not part of an objective function, so the carbon emissions could not be minimized. Baumgärtner et al. [29] calculated dynamic XEFs and MEFs for Germany and the year 2016 with an economic dispatch model and used them in the objective function of a multi-objective problem to design low-carbon L-MESs. They concluded that at the same costs, emissions can be reduced by 6% when using dynamic XEFs instead of annual XEFs and by up to 60% when using dynamic MEFs, which highlights the significance of using dynamic CEFs. In [30], optimization models considering DR have been reviewed. A lack of research on DR modeling for commercial and industrial consumers exists. Additionally, we strongly recommend the consideration of environmental effects by placing the carbon emissions in the objective function. In [31], dynamic XEFs and MEFs for 20 European countries were calculated, and a stylized price-based DR simulation was conducted. The results show that looking at national electricity systems, the effect of price-based DR on operational carbon emissions differs substantially from country to country and is dependent on the energy demand, generation mix, fuel costs, and carbon emission prices.

### *1.2. Energy System Optimization*

### 1.2.1. Multi-Objective Mixed Integer Linear Programming

We focus on multi-objective optimization, which combines two or more individual objective functions, e.g., the minimization of costs and carbon emissions, to gain a set of Pareto-optimal solutions [32]. These solutions are the best possible compromises and can be visualized as Pareto points on a scatter chart to give the user the option to choose from them.

The two most popular types of optimization methods are metaheuristics and mathematical programming [33]. Both types are used to optimize the operation and design of complex energy systems. Simplified, one can say that metaheuristics offer advantages for black-box models or for non-convex problems. In contrast to metaheuristics, mathematical programming can guarantee the achievement of a global optimal solution if an explicit equation-based model exists. Among the mathematical programming methods, mixed-integer linear programming (MILP) has shown to be effective for a wide range of cognate analyses. For instance, Zhang and Grossmann [34] listed 42 works with mathematical optimization models for industrial DR, among which, 35 were formulated as MILP models. For most energy-related components, the system behavior is nonlinear. However, the litera-

ture largely approximates this system behavior via MILP models where binary variables are used within the piecewise linear approximations of nonlinear system dynamics [35]. Besides the approximation through formulating a MILP for a nonlinear system, there are further complexity reduction methods, such as time series aggregation, that can be applied to MILP models for energy system optimization [36].

In this work, we chose MILP, since the formulated optimization problems can be solved with global optimality within reasonable time frames. We assume that an explicit equation-based model exists and that all occurring non-linear phenomena can be approximately modeled as piecewise linear phenomena.

### 1.2.2. Open-Source

In the scientific context, the open-source idea is also becoming increasingly prevalent in the field of energy system analysis [37]. Publishing source code under an open-source license that is approved by the Open Source Initiative and listed under https://opensource.org/licenses (accessed on 29 June 2022) not only increases transparency and reproducibility, but also the quality of the software due to the increased incentive for collaboration [38]. In contrast to models intended for large geographic scales, models for potential analyses and planning of L-MESs are often created by consulting firms, where the release of source code is often not in line with current corporate practices [39]. As a consequence, tools for the analysis of L-MESs are less often published as open-source.

### 1.2.3. Other Energy System Frameworks/Models

In recent years, the scientific community has produced some comprehensive energy system modeling frameworks. In [40], 75 modeling tools were reviewed. A review of 24 energy system models and model generators was presented in [41]. Kriechbaum et al. [42] reviewed open-source modeling frameworks for grid-based multi-energy systems. A review on the concepts and validation models of multi-energy systems was conducted in [43]. Based on the reviews above, Table 1 presents a comparison between DRAF and other existing frameworks and model generators capable of building bottom-up models for operation and investment decision support of L-MESs. Note that the table contains some full-fledged frameworks, such as oemof, whose wide range of functions and aspects cannot be presented here.

**Table 1.** Comparison of DRAF with other bottom-up model frameworks for operation and investment decision support of L-MESs divided by whether they are open-source. All links were last accessed on 25 November 2021. Sources: Based on [40] and own research as of 25 November 2021.

| Framework | Open-Source | Model | | | DRAF Focus | | | DRAF Features | | | | | |
|---|---|---|---|---|---|---|---|---|---|---|---|---|---|
| | | Methodology | MO | Purpose | DR | MTL | IPP | TSA | MG | PP | SG | IP | MD |
| EnergyPLAN (https://www.energyplan.eu) [44] | ✗ | Simulation | - | LTS/IDS | ✓ | ✗ | ✗ | ✗ | - | ✗ | ✓ | ✓ | ✓ |
| HOMER Pro (https://www.homerenergy.com) [45] | ✗ | Simulation [a] | - | IDS, ODS | ✓ | ✗ | ✗ | ✗ | - | (✓) | ✓ | ✓ | ✓ |
| TOP-Energy (https://www.top-energy.de) [46] | ✗ | Simulation+(MI)LP | ✓ | IDS, ODS | ✓ | ✓ | ✗ | ✓ | - | (✓) | ✓ | ✓ | ✓ |
| Calliope (https://www.callio.pe) [47] | ✓ (https://github.com/calliope-project/calliope) | (MI)LP | ✓ | IDS, ODS | ✓ | ✓ | ✗ | ✗ | ✓ | ✗ | ✗ | ✓ | ✗ |
| COMANDO (https://comando.readthedocs.io) [48] | ✓ (https://jugit.fz-juelich.de/iek-10/public/optimization/comando) | NLP | ✓ | LTS?, ODS | ✓ | ✓ | ✓ | ✗ | ✓ | ✗ | ✓ | ✗ | ✗ |
| ficus (https://ficus.readthedocs.io) [49] | ✓ (https://github.com/tum-ewk/ficus) | (MI)LP | ✗ | IDS, ODS | ✗ | ✗ | ✓ | ✗ | ✗ | ✗ | ✗ | ✓ | ✗ |
| oemof (https://oemof.readthedocs.io) [50] | ✓ (https://github.com/oemof/oemof) | (MI)LP | ✓ | LTS, IDS, ODS | ✓ | ✓ | ✓ | ✓ | ✓ | (✓) | ✓ | ✓ | ✗ |
| OpenTUMFlex (https://opentumflex.readthedocs.io) [51] | ✓ (https://github.com/tum-ewk/OpenTUMFlex) | (MI)LP | ✗ | IDS [b] | ✓ | ✗ | ✗ | ✓ | ✗ | ✗ | ✗ | ✗ | ✗ |
| OSeMOSYS (http://www.osemosys.org) [52] | ✓ (https://github.com/OSeMOSYS/OSeMOSYS) | LP | ✗ | IDS | ✗ | ✗ | ✗ [c] | ✗ | ✓ | ✗ | ✗ | ✓ | ✗ |
| Temoa (https://temoacloud.com) [53] | ✓ (https://github.com/TemoaProject/temoa) | LP | ✗ | LTS | ✗ | ✗ | ✗ | ✗ | ✗ | ✗ | ✓ | ✗ | ✗ |
| urbs (https://urbs.readthedocs.io) [54] | ✓ (https://github.com/tum-ens/urbs) | LP | ✗ | IDS, ODS | ✓ | ✗ | ✗ | ✗ | ✓ | ✗ | ✗ | ✓ | ✗ |
| **DRAF (our approach)** | ✓ (https://github.com/DrafProject/draf) | (MI)LP | ✓ | IDS, ODS | ✓ | ✓ | ✓ | ✓ | ✓ | ✓ | ✓ | ✓ | ✓ |

✓: Applicable; ✗: Not Applicable; (✓): Partly applicable; [a]: Optimization through automatic sensitivity analysis; [b]: focus on residential sector; [c]: focus on developing countries; Simulation: computer simulation; LTS: long term scenarios; IDS/ODS: investment/operation decision support; MO: multi-objective; TSA: time series analysis; MG: model generator; PP: parameter preparation; SG: scenario generator; IP: interactive plotting; DR: demand response; MTL: multiple temperature levels; IPP: industrial production processes; MD: metadata handling.

A powerful and versatile framework is the open energy modeling framework (oemof) [55]. More specifically, oemof is an organizational framework that bundles software for energy (system) modeling, such as the model generator oemof-solph [56]. However, oemof-solph uses Pyomo for the model generation, which builds models slower than the Gurobi Python interface GurobiPy [57]. Metadata such as units, parameter descriptions, and parameter value sources are not handled by the framework. While oemof cosmos offers several packages for parameter preparation, such as the load curve generation package demandlib [58], the preparation of important data for the modeling of price-based DR in L-MESs, such as dynamic CEFs, is not included. Very recently, Langiu et al., proposed the framework for component-oriented modeling and optimization for nonlinear design and operation (COMANDO) [48]. It focuses on nonlinear optimization; however, parameter preprocessing, interactive plotting, and metadata handling are not included in the package. There are also efforts to exploit the speed of the relatively new Julia programming language within energy system modeling, e.g., next energy modeling system for optimization (NEMO) [59], which was informed by the open source energy modelling system (OSeMOSYS). However, Julia is still a young programming language that lacks the massive community support needed for our target group, namely, programming beginners [60].

Some frameworks can be used in a flexible manner for manifold analyses (temporal and geographic resolutions) due to their modular structure and the clear separation between program logic and data, such as oemof [50]. This generic approach offers crucial advantages for large-scale power system analyses in terms of transparency, reusability, and maintainability. However, in order for the software to be used as decision support in an L-MES, it must be adapted to the specific problem with the following additional steps:

- Model adaptation to industry-specific conditions;
- Research and processing of market data, such as dynamic CEFs (depending on the country, year, and temporal resolution) and cost functions;

- Generation of weather-dependent energy-relevant time series, such as energy yield time series for photovoltaics (PVs) or thermal load profiles;
- Preparation, analysis, and plausibility checking of project-specific data, such as electrical load profiles,
- Model parameterization;
- Adaptation of result output functions, such as plots and tables to the particular data structure.

Commercial tools, such as the first three in Table 1, have addressed this need and can speed up the modeling process by specializing in specific application areas. A corresponding open-source alternative that leverages the potentials of the open-source idea (see Section 1.2.2) is currently not available.

### 1.3. Contributions

The main contributions of this paper are the development and demonstration of DRAF, an easy-to-use open-source Python decision support framework for optimizing DR-related design and operation of L-MESs. DRAF uses multi-objective MILP optimization to enable the user to quantify the cost and carbon emission reduction potential of existing and future flexibility options of L-MESs. The target groups of DRAF are researchers in the energy field and decision-makers in the commercial and industrial sectors. The source code of DRAF can be found at https://github.com/DrafProject/draf [61] (accessed on 28 June 2022). A secondary contribution of this study is the concise review of the current state of the art in open-source energy optimization software for DR.

The remainder of this paper is organized as follows. First, the most important elements and components of DRAF are described in Section 2, while referring to the extensive appendix with screenshots (Appendix A) and component templates (Appendix B). Second, the application of DRAF to three different simplified real-world case studies is demonstrated in Section 3. This is followed by a discussion, conclusion, and future research in Section 4.

## 2. The Demand Response Analysis Framework (DRAF)

### 2.1. Overview

Figure 1 shows the main functional elements of DRAF. One can see that DRAF provides a toolbox for every step of typical energy system analysis and the optimization process of an L-MES decision maker. More specifically, DRAF is designed to answer the three questions illustrated in Figure 2.

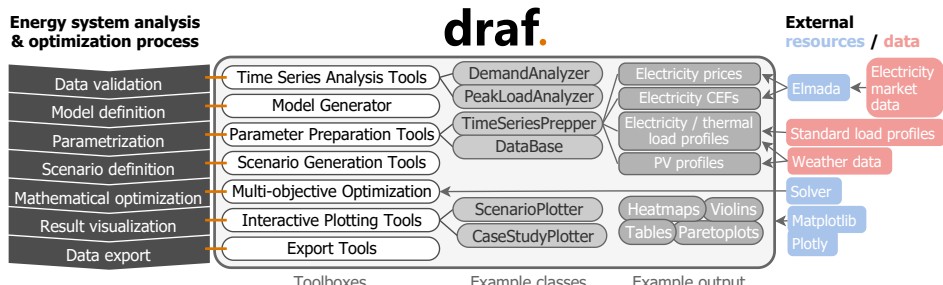

**Figure 1.** Schematic depiction of DRAF's toolboxes, including examples and how they relate to the energy system analysis and optimization process and the used external resources/data. For example, parameterization is supported by DRAF through the parameter preparation tool TimeSeriesPrepper, which, e.g., provides electricity prices using the tool Elmada as an external resource.

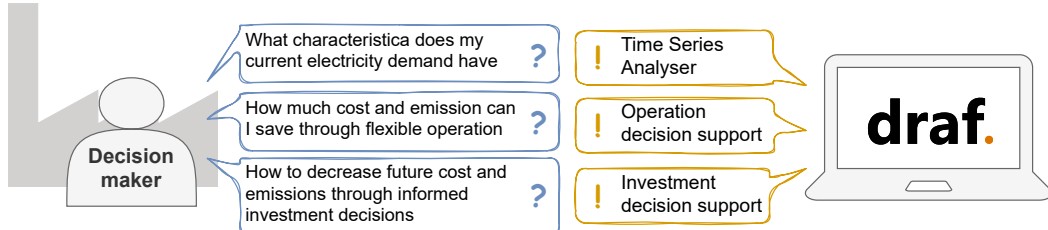

**Figure 2.** Key questions answered by DRAF.

The general architecture of DRAF is presented in Figure 3. A typical use case is described in the following. First, a user imports and analyzes time series data of, e.g., electricity and natural gas historically purchased by the analyzed L-MES operator using DemandAnalyzer. Informed by the findings of the analyses, the user then instantiates a CaseStudy object of the desired analysis year and the country/address/coordinates of the L-MES. Subsequently, a first reference scenario that includes a model is added to the case study using component templates and the model generator. While DataBase is used here to provide and describe default parameters, the TimeSeriesPrepper prepares relevant time series, such as the day-ahead market prices, dynamic CEFs, or PV profiles. Different scenarios are then added by duplicating the reference scenario and changing specific parameters. After the model is solved by an external MILP solver, the results are stored in the cenario object. Finally, all parameters and results can be visualized either for each scenario (ScenarioPlotter) or for all scenarios in the case study (CaseStudyPlotter). The interconnected classes and modules allow for a fluent, explorative analysis process using the dot operator. E.g., `cs.scens` returns an overview of all defined scenarios; `cs.scens.sc2.res.P_PV_fi_T.plot()` plots the feed-in PV power of the scenario sc2. DRAF handles metadata, i.e., parameters can be stored together with descriptions, units, and sources. This motivates the input of metadata which can be used in plotting and exporting, prevents misunderstandings, and helps to document the meaning of an optimization model.

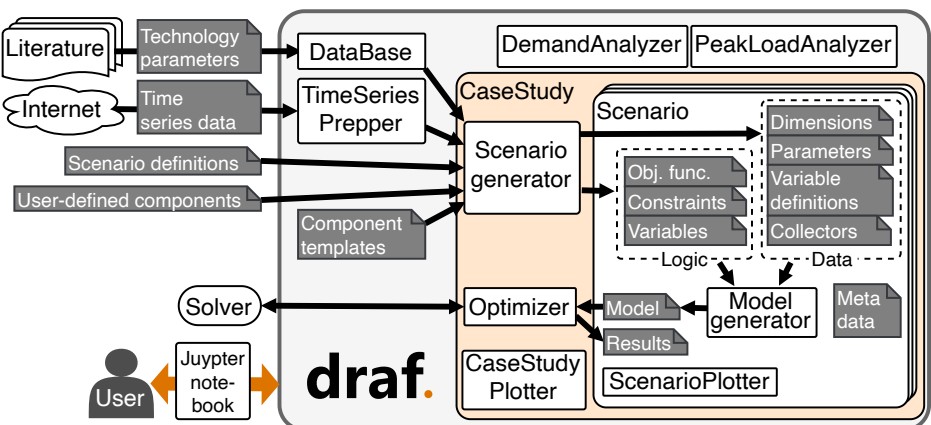

**Figure 3.** Software architecture of DRAF. The user interacts with DRAF via Jupyter notebooks. To carry out an analysis, the user initiates a CaseStudy and defines scenarios using the scenario generator, component templates, and parameter preparation tools (DataBase and TimeSeriesPrepper). The model generator builds optimization models from the model definition consisting of data and logic. After the optimization, the results can be plotted either via CaseStudyPlotter or via ScenarioPlotter using convenient dot notation (e.g., `cs.plot.tables()`; see Figure A3).

There are other energy modeling frameworks that also consider flexible demand. However, they do not focus on the modeling of L-MESs, such as individual commercial or industrial DERs. Since they are very generic, they technically allow the modeling of individual industrial sites; however, this abstraction comes with the cost of higher computational complexity and increased familiarization time with the tool for the user.

Additionally, they do not provide easy access to necessary market information, such as wholesale prices, dynamic emission factors, and technology investment costs. Versatility and portability are important aspects of DRAF. Both are ensured through a modular structure of DRAF that can be used as a generalized framework that can easily be adapted to specific applications.

### 2.2. Python as a High Level Programming Language

For the development of DRAF, the open-source, general-purpose programming language Python [62] 3.9 was chosen. A general-purpose programming language allows all steps of the analysis process from market data acquisition to data preparation, model building, scenario definition, and result visualization to be defined and reproducibly documented in a single environment. Since 2010, Python became the standard for new energy system models [41]. At the time of writing, Python is the most popular easy-to-use high-level programming language [63]. It has rich libraries for data handling and visualization (e.g., Pandas [64], Matplotlib [65], and Plotly [66]). The open-source optimization modeling language Pyomo [67] provides access to important MILP solvers, such as Gurobi [68]. However, most importantly, Python has a vibrant community that supplies help and solutions to almost any problem that might occur.

### 2.3. Time Series Analysis Tools

As can be seen in Figure 1, the Time Series Analysis Tools refer to a DRAF toolbox. The included tools, DemandAnalyzer and PeakLoadAnalyzer, are described in the following.

#### 2.3.1. DemandAnalyzer

Often, DR analyses are based on a time series of historic energy demands, e.g., the electricity demand of the year before. Before starting the modeling process, an analysis of this time series is helpful to validate the correct time-series length and to get key metrics, such as the peak-to-average ratio, load percentiles, and usage patterns. DRAF provides such an analysis with `DemandAnalyzer`. Figure A1 shows a screenshot of an example time series analysis.

#### 2.3.2. PeakLoadAnalyzer

Based on the `DemandAnalyzer` object, the `PeakLoadAnalyzer` can be used; see screenshot in Figure A2. It shows the peak loads above a user-defined threshold and the cost reduction potential that originates from a given peak load price.

### 2.4. Parameter Preparation Tools

The Parameter Preparation Tools is a toolbox (cf. Figure 1) that supports the preparation of parameters for the optimization model.

#### 2.4.1. TimeSeriesPrepper

DRAF's TimeSeriesPrepper allows the user to prepare time series such as the ambient air temperature; renewable electricity generation profiles; CEFs; and day-ahead market prices from basic input data, such as an address that converts to geographic coordinates, the analyzed year, and the time step width.

#### Carbon Emission Factors (CEFs) and Electricity Prices

In DRAF, CEFs and day-ahead prices for most European national electricity systems are automatically calculated for the given year and frequency with the open-source tool Elmada [69], as described in [31]. The latest Elmada version, v0.1.0 [70], supplies hourly and quarter-hourly time series for 30 European countries, mainly using data from the European Network of Transmission System Operators for Electricity (ENTSO-E) transparency platform [71]. For the CEFs, the user can choose between XEFs and MEFs, depending on the analysis question. For the electricity prices, the user can choose between day-ahead

spot market prices which are referred to as RTP, time-of-use (TOU) pricing, and a flat price. In TOU, if not stated otherwise, time steps are grouped into high price and low price times; high price times apply between 8 a.m. to 8 p.m. during workdays. The high/low price is defined by the mean of RTP during high/low price times, respectively. The flat price is the annual mean of the RTPs.

### Photovoltaic Power Profiles

If the user does not provide PV profiles for the analyzed location, DRAF uses the Global Solar Energy Estimator (GSEE) [72] to generate PV profiles from the geographic coordinates (or a valid address), global and diffuse radiation, and ambient temperature time series. For Germany only, DRAF checks for available weather data from the nearest weather station from [73] and downloads it in the background. This functionality may be extended to other countries as data become available.

### Electrical and Thermal Load Profiles

The TimeSeriesPrepper module of DRAF also provides functions to create electrical and thermal load time series. In the electrical case, standard load profiles from [74] are used while considering public holidays from the given region with the Python holidays package [75]. For thermal load profiles, ambient temperature data from [73] are used to approximate heating and cooling demand time series.

### 2.4.2. DataBase

The reasoning behind the DataBase is to pragmatically provide the user with an expandable library of technical and market-related parameters, together with metadata such as units, descriptions, and scientific sources. Some of the values are used in the component template definitions in Appendix B.

### 2.5. Component-Based Model Generator

To avoid code repetition, and to make DRAF and the code written by the user maintainable, extensible, and adaptable, the toolbox Model Generator (cf. Figure 1) is implemented that creates model constructs in a lazy fashion; see Algorithm 1. Furthermore, the model generator keeps the model as light as possible, creating only the dimensions, parameters, optimization variables, and constraints that are needed. This allows the provision of adaptable and extensible component templates without limiting the user's freedom to build any other MILP model.

A component class consists of the two functions that define the component, `param_func` and `model_func`. The `param_func` defines dimensions, parameters, variables, and collectors for a given scenario. The `model_func` later uses these objects to build constraints and to connect the component to other components by contributing linear expressions to their collectors. The listings in Figure 4 show examples of these functions. The first listing defines a simple PV component. Note that dimensions and collectors are not needed for this simple PV component. The second listing shows relevant parts of the Main component, which defines general relationships that do not originate from a specific technical component.

For adding constraints and objective functions, the user can choose between Pyomo and GurobiPy syntax. While Pyomo supports different solvers, GurobiPy is limited to the commercial solver Gurobi but builds models with less computational effort.

Component interdependencies are considered using so-called collectors, which collect linear expressions across components and aggregate them in another component. For example, the collectors for investment costs are defined in the Main component with `sc.collector("C_inv_")`; then, the components PV and fuel cell (FC) contribute to it with `c.C_inv_["PV"] = ...` and `c.C_inv_["FC"] = ...`. Finally, the Main component uses the collector to aggregate the investment costs with `sum(c.C_inv_.values())`. Collectors can collect scalar values, e.g, to aggregate investment costs of different components to total costs, or collect functions to access multi-dimensional vectors, e.g., to build an electricity

balance for each time step; see Figure 5. If a component uses a collector, the constraints of that component must be built after the constraints of all components that contribute to that collector. This dependency restricts the order of submodel creation, which is resolved by executing a topological sort. This makes components reusable, so the user can conveniently choose from different storage and conversion technology components and modeling options, such as the consideration of investments or minimal part-load behavior, without inflating the model with overhead constructs. The user defines optimization models by using component templates (see Appendix B) and/or self-written technology components.

---

**Algorithm 1:** Model generation.

**Input** : Case study configuration $\Omega$ including year, frequency, modeling horizon, geo coordinates, if investment is considered, etc.; User-selected components $C$; Scenario definitions $D$

**Output**: Case study $S$ including scenarios and parametrized optimization models

1 Initialize case study $S$ with configuration $\Omega$

  // Build scenarios with defined parameters and variables:

2 **for** each scenario definition $d$ in $D$ **do**

3      **if** $d$ is based on a base scenario **then**

4          Initialize scenario $s$ as a copy of the respective base scenario

5      **else**

6          Initialize a new scenario $s$

7      Define and/or update parameters and variables in $s$ using $C$, $\Omega$, and $d$

8      Add $s$ to case study $S$

  // Build optimization model for each scenario:

9 Sort components $C$ topologically considering component interdependencies

10 **for** each scenario $s$ in case study $S$ **do**

11      Initialize optimization model $m$ and add to scenario $s$

12      Initialize optimization variables from variable definitions of $s$

13      **for** each component $c$ in $C$ **do**

14          Set objective function and constraints of component $c$ to model $m$ using parameters, optimization variables and collectors of $s$

15          Register linear expressions to collectors

---

```python
class PV(Component):

 def param_func(self, sc:Scenario):
   # sc.dim(...)    (Dimensions could be defined here if applicable)
   sc.param("c_PV_inv_", data=460, doc="Inv. costs", unit="EUR/kW_peak")
   sc.param("P_PV_max_", data=500, doc="Max. peak power", unit="kw_peak")
   sc.param("P_PV_profile_T", data=getProfile(), unit="kW_el",
            doc="Elec. power of 1kW_peak per time step")
   sc.var("P_PV_CAPn_", doc="New capacity", unit="kW_peak")
   sc.var("P_PV_OC_T", doc="Own consumption", unit="kW_el")
   sc.var("P_PV_FI_T", doc="Feed-in", unit="kW_el")
   # sc.collector(...)    (Collectors could be defined here if applicable)

 def model_func(self, sc, m:Model, d:Dims, p:Params, v:Vars, c:Collectors):
   m.addConstr(v.P_PV_CAPn_ <= p.P_PV_max_)
   m.addConstrs((p.P_PV_profile_T[t] * v.P_PV_CAPn_
                 == v.P_PV_FI_T[t] + v.P_PV_OC_T[t]
                 for t in d.T))

   # The PV component submits to three different collectors:
   c.C_inv_["PV"] = v.P_PV_CAPn_ * p.c_PV_inv_
   c.P_EL_source_T["PV"] = lambda t: v.P_PV_OC_T[t]
   c.P_EG_feedin_T["PV"] = lambda t: v.P_PV_FI_T[t]
```

**Figure 4.** *Cont.*

```
class Main(Component):

 def param_func(self, sc:Scenario):
   sc.collector("C_inv_", doc="Total investment costs", unit="EUR")
   sc.collector("P_EL_source_T", doc="Power sources", unit="kW_el")
   sc.collector("P_EL_sink_T", doc="Power sinks", unit="kW_el")
   ...

 def model_func(self, sc, m:Model, d:Dims, p:Params, v:Vars, c:Collectors):
   # Investment cost aggregation using a collector:
   m.addConstr(v.C_inv_ == sum(c.C_inv_.values()))

   # Electricity balance using two collectors:
   m.addConstrs((sum(f(t) for f in c.P_EL_source_T.values())
               == sum(f(t) for f in c.P_EL_sink_T.values())
               for t in d.T))
   ...
```

**Figure 4.** Listings of example components.

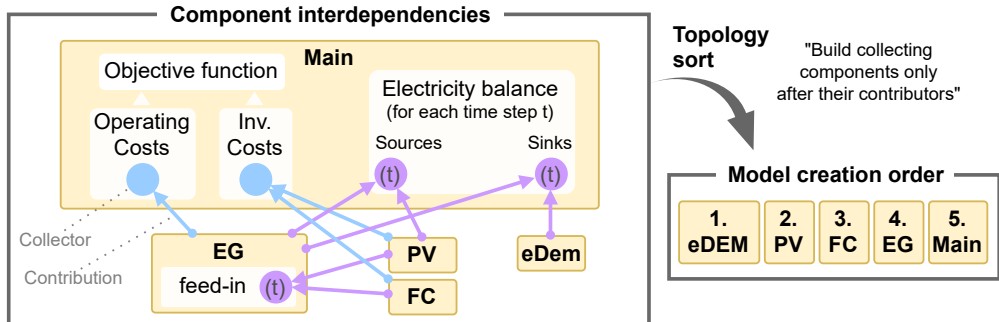

**Figure 5.** An example where collectors (circles) induce component interdependencies. The Main component collects operating and investment costs and the two sides of the electricity balance for each time step. The electricity grid (EG) component collects for each time step electricity that is fed into the grid—here, the contributors are the fuel cell (FC) and the photovoltaic system (PV). This separate feed-in collector is important for the reusability of the EG component. The electricity demand (eDem) only affects the electricity balance sinks. The component interdependencies are considered within model generation through topology sort. This ensures that all submodels contributing to a collector are built before that collector is executed.

## 2.6. Component Templates

DRAF provides a set of component templates so that users do not need to start from scratch. The model is assembled from the Main component and multiple technology components. The Main component includes the objective function and all general sets, parameters, variables, and balances. Technology components are, e.g., energy demands, conversion and storage technologies, and interfaces to external entities, such as the electricity grid.

### 2.6.1. The Component Template Main

The Main component is a special component template that consists of the definition of the objective function and general balances and constraints. The Main component yields a deterministic multi-objective combined design and operation problem, which is described in the following. The formulation is partly based on previous work [76,77]. By default, DRAF works with equidistant time steps and assumes perfect foresight. The temporal resolution is defined by the user and not restricted; however, hourly or quarter-hourly resolution is required if the functionality of the TimeSeriesPrepper module is used; see Section 2.4.1. The default modeling horizon is one year, but can be customized. General sets are discrete time steps $t \in \mathcal{T} := \{t_1, \ldots, t_{|\mathcal{T}|}\}$, components $j \in \mathcal{J} := \{\text{eDem, EG, PV, BES}, \ldots\}$, and flow types $i \in \mathcal{I} := \{\text{electricity, heat 1, heat 2, cool 1, cool 2, product 1}, \ldots\}$. In the follow-

ing, optimization variables are denoted with **bold** symbols and are non-negative continuous variables unless stated differently.

The objective function of the MILP is to minimize the weighted sum of the total annualized cost (TAC) $\boldsymbol{C}^{\text{tot}}$ and the annual carbon emissions $\boldsymbol{CE}^{\text{tot}}$:

$$\text{minimize} \quad \boldsymbol{Z} := (1 - \alpha)\pi^{\text{C}}\boldsymbol{C}^{\text{tot}} + \alpha\pi^{\text{CE}}\boldsymbol{CE}^{\text{tot}} + \sum_j \boldsymbol{X}_j^{\text{penalty}} \tag{1}$$

where $\alpha$ is the Pareto weighting factor $\in [0..1]$ and $\pi^{\text{C}}$, $\pi^{\text{CE}}$ are the Pareto normalization factors which can be identified by a simple algorithm to enable a more even distribution of the Pareto points; cf. Figure 14. $\boldsymbol{X}_j^{\text{penalty}}$ is a general penalty term which is only used in rare cases when there is an incentive that is not related to costs or emissions, e.g., to model uncontrolled battery electric vehicle (BEV) charging (in Appendix B.6), where the incentive is to charge based on time constraints.

The annualized total costs $\boldsymbol{C}^{\text{tot}}$ are the sums of the annual operating costs, the annualized investment costs, and the maintenance costs of all components $j \in \mathcal{J}$:

$$C^{\text{tot}} = \underbrace{\sum_j C_j^{\text{op}}}_{\text{operation costs}} + \underbrace{\sum_j C_j^{\text{invAnn}}}_{\text{annualized investment costs}} + \underbrace{\sum_j C_j^{\text{rmi}}}_{\text{maintenance costs}} \tag{2}$$

$$C^{\text{tot}} \in \mathbb{R}, \quad C_j^{\text{op}} \in \mathbb{R}^{|J|}$$

The total carbon emissions $\boldsymbol{CE}^{\text{tot}}$ are the sums of yearly operating carbon emissions of all components $j \in \mathcal{J}$:

$$\boldsymbol{CE}^{\text{tot}} = \sum_j \boldsymbol{CE}_j \tag{3}$$

$$\boldsymbol{CE}^{\text{tot}} \in \mathbb{R}, \quad \boldsymbol{CE}_j \in \mathbb{R}^{|J|}$$

Time-step average flow values, such as electrical power, the thermal energy flow of a specific temperature level, or a product flow, are generically denoted with $\boldsymbol{\Phi}_{i,j,t}$. Balances are defined by equating the sum of all input flows $\boldsymbol{\Phi}_{i,j,t}^{\text{source}}$ with the sum of all output flows $\boldsymbol{\Phi}_{i,j,t}^{\text{sink}}$, for all flow types $i \in \mathcal{I}$:

$$\sum_j \boldsymbol{\Phi}_{i,j,t}^{\text{source}} = \sum_j \boldsymbol{\Phi}_{i,j,t}^{\text{sink}} \quad \forall t \in \mathcal{T}, i \in \mathcal{I} \tag{4}$$

2.6.2. Technology Component Templates

Appendix B contains the mathematical formulation of all component templates. Note that besides conversion and storage technologies, demands, and market interfaces of commodities, such as electricity (Appendix B.1) or fuels (Appendix B.2), are also individual component templates.

Most of the collectors exist within the Main component templates, and a few in others; e.g., in the EG component (Appendix B.1):

$$\boldsymbol{P}_t^{\text{eg,sell}} = \sum_j \boldsymbol{P}_{t,j}^{\text{sell}} \tag{5}$$

or in the Fuel component (Appendix B.2):

$$\boldsymbol{F}_f^{\text{fuel}} = \sum_j \boldsymbol{F}_{f,j} \quad \forall f \in \mathcal{F} \tag{6}$$

The total annualized investment costs $C_j^{\text{invAnn}}$ and the maintenance costs $C_j^{\text{rmi}}$ are usually defined within storage and conversion components:

$$C_j^{\text{invAnn}} = k^{j,\text{af}}(r, N^j) c^{j,\text{inv}} P^{j,\text{capn}} \tag{7}$$

$$C_j^{\text{rmi}} = k^{j,\text{rmi}} c^{j,\text{inv}} P^{j,\text{capn}} \tag{8}$$

where $j$ stands for the component $j$, $c^{j,\text{inv}}$ are the specific investment costs, and $P^{j,\text{capn}}$ the new capacities. $k_j^{\text{af}}(r, N^j)$ are the component-specific annuity factors defined in Equation (9) following, e.g., [78], where $r$ is the calculated interest rate and $N^j$ is the operation life in years for component $j$.

$$k_j^{\text{af}}(r, N^j) = \frac{r(1+r)^{N^j}}{(1+r)^{N^j} - 1} \tag{9}$$

For the component templates, additional sets are defined: Fuel types $f \in \mathcal{F} := \{\text{biogas, naturalgas}\}$, condensing temperature levels $c \in \mathcal{C} := \{1, \ldots, |C|\}$, heating temperature levels $h \in \mathcal{H} := \{1, \ldots, |H|\}$, cooling temperature levels $n \in \mathcal{N} := \{1, \ldots, |N|\}$, and thermal demand temperature levels $l \in \mathcal{L} := \mathcal{H} \cup \mathcal{N}$.

### 2.7. Scenario Generation and Optimization

The scenario generator in DRAF provides convenient scenario generation. Scenarios can either be created manually or created in batches. For manual scenario creation, an existing object is cloned with `sc = cs.add_scen(based_on=<scen_id>)`, whose parameters can be subsequently updated with `sc.update_params(param1=value1, param2=value2, ...)`; see also Algorithm 1. The batch scenario creation using `cs.add_scens()` can be seen as sensitivity analysis, which automatically creates a scenario for each combination of given parameters and parameter values. This is useful, e.g., for optimizing the system for different energy and carbon emission prices. When solving optimization models for a case study, the user can choose to solve the scenarios in parallel (`cs.optimize(parallel=True)`) using the distributed execution framework Ray [79] or serially to rely on the parallelization of the solver.

### 2.8. Visualization

DRAF provides a rich interactive visualization toolbox built into the `CaseStudy` and `Scenario` classes (cf. Figure 1). The dot notation allows convenient plotting since the data and metadata are internally fetched. E.g., after optimizing multiple scenarios, `cs.plot.pareto()` plots the Pareto front of all scenarios in the case study, similarly to Figure 14, and `cs.scens.sc3.plot.sankey()` plots the Sankey diagram of scenario sc3; see Figure A4. Interactive parameter and result exploration are available thanks to the diverse capabilities of Ipython [80] and Plotly [66]; see also Figure A5.

## 3. Case Studies

In the following, DRAF's features are demonstrated in three case studies that we consider to be of interest to the reader. The case studies are based on real companies in southern Germany with modified values for data protection reasons. In Case Study 1, the production schedule of a cement plant is optimized considering price-based DR. This case study was selected as it illustrates DRAF's support for flexible industrial production processes. In Case Study 2, the design of a battery energy storage (BES) and a PV system at an industrial site is optimized considering multiple flexibility applications and differentiating between existing and new technologies. Case Study 3 covers a more sophisticated superstructure for a greenfield L-MES. This last case study demonstrates multi-objective optimization, the value of the scenario generator, and the consideration of multiple temperature levels. The code for the presented case studies is available at https://github.com/DrafProject/draf_demo_case_studies (accessed on 28 June 2022). The calculations were performed using DRAF v.0.2.0 [61].

### 3.1. Case Study 1: Price-Based DR Potential of an Industrial Production Process

In the first case study, which is based on the previous work [81,82], we apply DRAF to the problem of quantifying the cost and carbon emission reduction potential of price-based DR of a cement milling process.

The setup of the case study is shown in Figure 6. In it, there are two electric cement mills that turn cement clinker into three different cement sorts which are stored in separate silos to serve a given cement demand. For each time step, the cement mills can either be powered down or produce one compatible cement sort in full-load with a sort and machine-specific production efficiency; see Figure 7 left. Machine 1 is compatible with sorts 1 and 3, and machine 2 is compatible with sorts 2 and 3; see Figure 6. The cement clinker supply is not a bottleneck in the production process, so it was modeled as unrestricted. The cement demand was generated by breaking down the total monthly demand into the working hours of the plant; see Figure 7 right. A MILP model was built using DRAF's component-templates Main, EG, pDEM, PP, and PS (see Appendices B.1,B.12–B.14) to minimize the TAC $C^{tot}$ of the system for 8760 hourly time steps from the year 2019. Therein, cement mills are represented by machines, cement clinker by raw material, and cement silos by product storage. Dynamic TOU and RTP pricing schemes and XEFs were prepared by DRAF's TimeSeriesPrepper, described in Section 2.4.1. The peak power price was €50 kW$_P^{-1}$. A standard load profile for continuous production was scaled to the annual energy of 5 GWh and used as fixed electricity demand. The cement mills have nominal capacities of 3.5 MW each and a minimum part-load factor of 1. Each machine start-up and sort-change costs €10.00 due to the inefficient operation associated with it. Cement mill 1 was unavailable due to revisions from the 15th to 16th of March, and the cement mill 2 from the 15th to 16th of February. Each cement silo had a capacity and initial filling of 5 kt and a minimum filling level of 1 kt. Since part-load operation was not possible, the deviation between the last and the initial filling level was evaluated with a factor $k$ and penalized via the operating costs. $k$ was composed of the worst efficiency and three times the average electricity price of the year. Thus, the deviation was minimized without introducing infeasibility. For brevity, investment in storage extension was not allowed, even though this would be possible with the model and would be an interesting analysis.

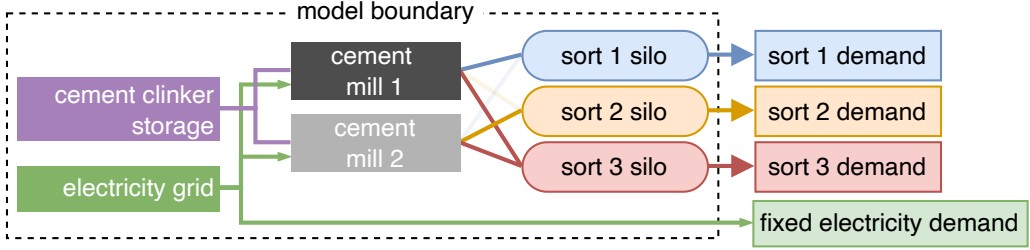

**Figure 6.** Setup of cement case study.

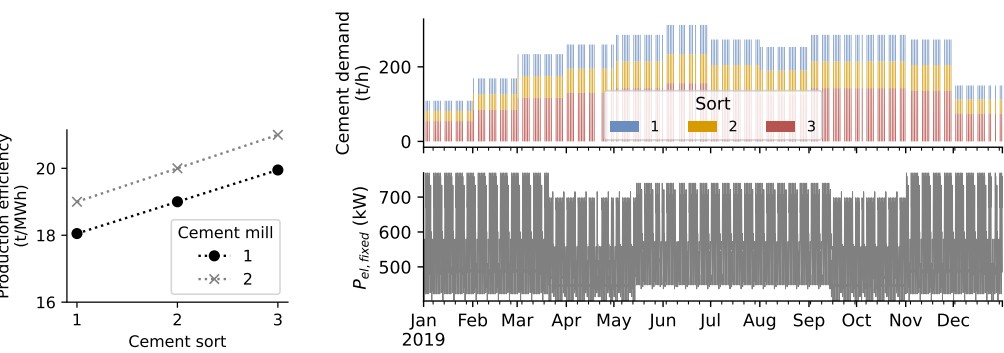

**Figure 7.** Left: Machine and cement-sort specific production efficiencies. Right: Cement demand and fixed electricity demand.

Despite the technical constraints and the small price spreads of 2019 the load shifting led to savings of €149,000, (1.9% of electricity costs) and 1.4 kt$_{CO2eq}$ (9.2% of operating carbon emissions) per year. However, these are theoretical values, as complete foresight was assumed. The results for ten of the 365 optimized days are shown in Figure 8. The plots show that while the silo filling levels look similar, the scheduling of the cement sorts between the two scenarios differs substantially on an hourly basis. The electricity price is the main factor in the production decision. Exceptions are due to sort-switching and start-up costs. The Pearson correlation coefficient *r* between the electricity price and the purchased electricity for the whole year is −0.29 with a TOU pricing scheme and −0.58 with RTP; see Figure 9. Since machine 1 is more efficient than machine 2, the most energy-intensive sort 1 is only produced on machine 1. Sort 1 is only produced on machine 1 and sort 2 only on machine 2. The electrical peak demand was not lowered.

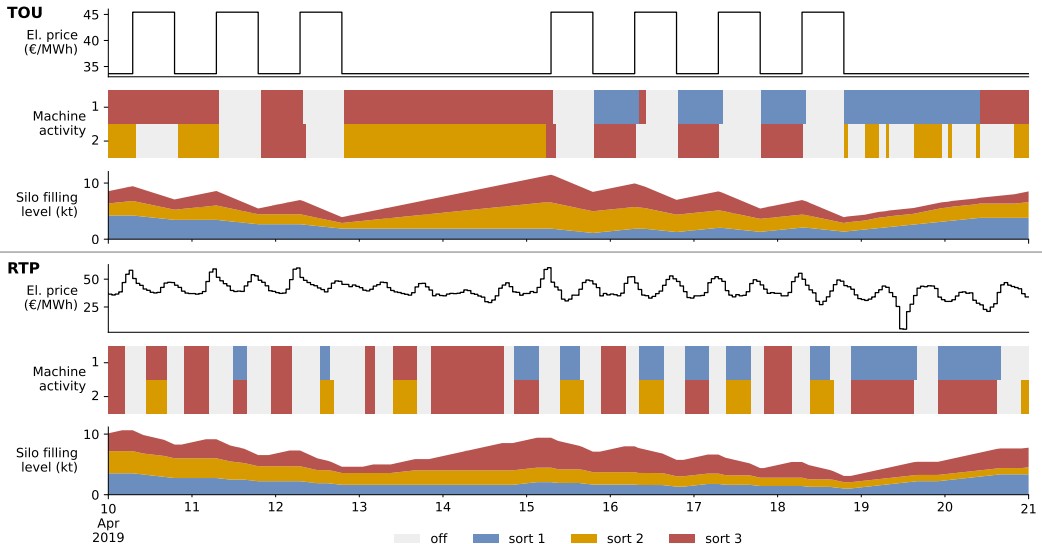

**Figure 8.** Results of Case Study 1: Resulting production plans, price schemes, and silo filling levels for ten sample days in April. Top: Reference scenario with the time-of-use pricing scheme. Bottom: Scenario with hourly German day-ahead market prices.

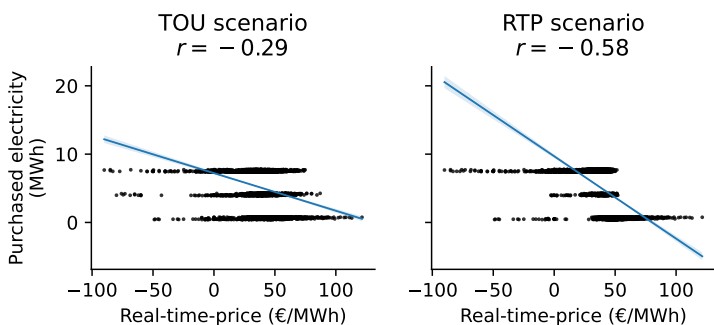

**Figure 9.** Correlations between purchased electricity and real-time-prices for both scenarios.

### 3.2. Case Study 2: Design Optimization of a Multi-Use BES and PV System

Figure 10 shows the setup and problem of Case Study 2.

For the inflexible electricity demand, anonymized real industrial data of the year 2020 were used, which are analyzed in Figure A1. Further input parameters were dynamic day-ahead market prices plus €62.3 MWh$^{-1}$ electricity taxes and levies and €100 kW$^{-1}$ peak electricity price. Specific investment cost forecasts for 2022 for PV and BES were taken from [83] with the values €384 kW$_p^{-1}$ and €209 kWh$^{-1}$, respectively.

Four scenarios were modeled: REF (reference scenario), optBES (allows BES), optPV (allows new PV), and optBesPv (allows both). Tables 2 and 3 and Figure A3 show the results of Case Study 2. Figure 11 shows the resulting electricity balance and the RTP of scenario optBesPv for one exemplary week (Monday–Sunday).

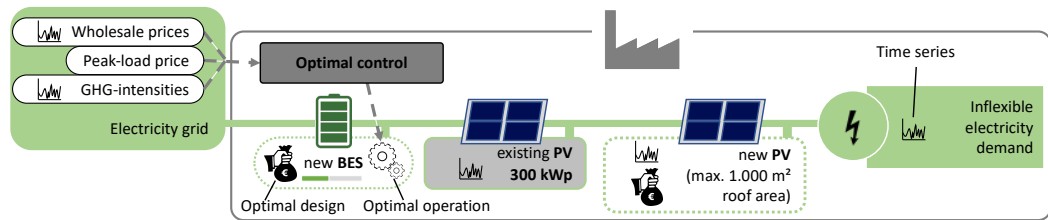

**Figure 10.** Setup and problem of Case Study 2. **Setup**: A company that can buy electricity from the grid and sell it to the grid has an existing $300\,\text{kW}_{\text{peak}}$ PV system, $1000\,\text{m}^2$ additionally available rooftop space for the installation of a new PV system, and an inflexible electricity demand. **Problem**: The design (nominal capacity/power) of the BES and the new PV is to be optimized assuming optimal charging and discharging of the BES considering peak shaving, RTPs through hourly wholesale prices, and the optimization of self-consumption.

**Table 2.** Capacities and investment costs of Case Study 2.

| Scenario | CAPx | | CAPn | | $C^{\text{inv}}$ [k€] | | $C^{\text{inv,ann}}$ [k€/a] | |
|---|---|---|---|---|---|---|---|---|
| | **BES** | **PV** | **BES** | **PV** | **BES** | **PV** | **BES** | **PV** |
| REF | 0 | 300 | 0.0 | 0.0 | 0.0 | 0.0 | 0.0 | 0.0 |
| optBes | 0 | 300 | 233.4 | 0.0 | 48.8 | 0.0 | 4.3 | 0.0 |
| optPV | 0 | 300 | 0.0 | 153.8 | 0.0 | 59.1 | 0.0 | 4.6 |
| optBesPv | 0 | 300 | 265.6 | 153.8 | 55.5 | 59.1 | 4.8 | 4.6 |

**Table 3.** Peak reductions of Case Study 2.

| Scenario | $P^{\text{max}}$ | $P^{\text{max,reduction}}$ | | $W^{\text{buy}}$ | $W^{\text{sell}}$ |
|---|---|---|---|---|---|
| REF | 1445 kW | 0 kW | 0.0% | 7.122 GWh/a | 0.000 GWh/a |
| optBes | 1330 kW | 115 kW | 0.1% | 7.130 GWh/a | 0.000 GWh/a |
| optPV | 1445 kW | 0 kW | 0.0% | 6.952 GWh/a | 0.000 GWh/a |
| optBesPv | 1320 kW | 125 kW | 0.1% | 6.960 GWh/a | 0.000 GWh/a |

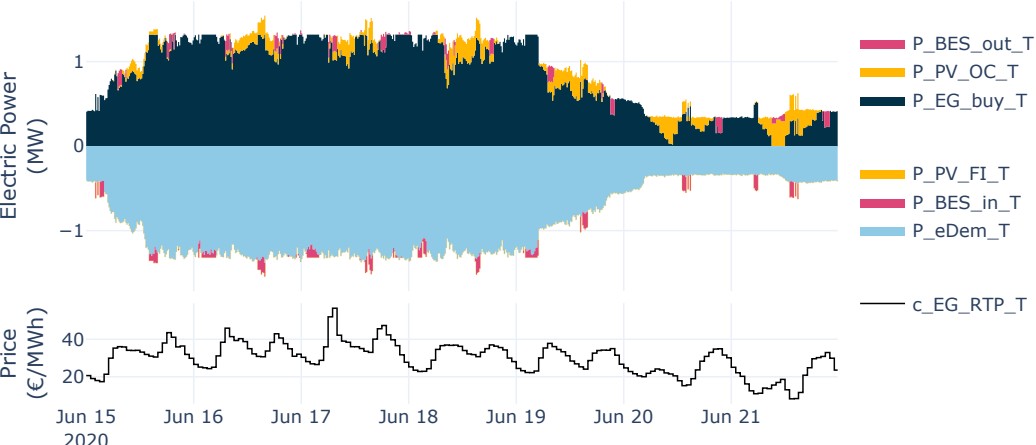

**Figure 11.** Top: Stacked area plot of electricity balance. Energy sources are positive. Energy usages are negative. Bottom: Real-time prices.

### 3.3. Case Study 3: Multi-Objective Design and Operational Optimization of Thermal-Electric Sector Coupling

The third case study demonstrates the optimization of the design and operation of a more sophisticated industrial L-MES. This time it is a greenfield project—i.e., there are no existing technologies. Besides the inflexible electricity demand, the L-MES incorporates cooling and heating demands at different temperature levels. Figure 12 shows an overview and the superstructure of the analyzed L-MES. Up to 20 MWh/h of electricity can be bought from and sold to the grid with hourly prices and XEFs. Assuming plant-wide optimization with perfect foresight of XEFs, electricity prices, and PV yield profiles, Pareto-optimal design and operational configurations were to be identified that fulfilled the thermal and electrical energy demands. The following component templates from Appendix B were used: cDem, hDem, eDem, EG, Fuel, PV, BES, CHP, HOB, HP, P2H, H2H1, TES, and Main. The scheme in Figure 13 provides details on the modeling of the different temperature levels.

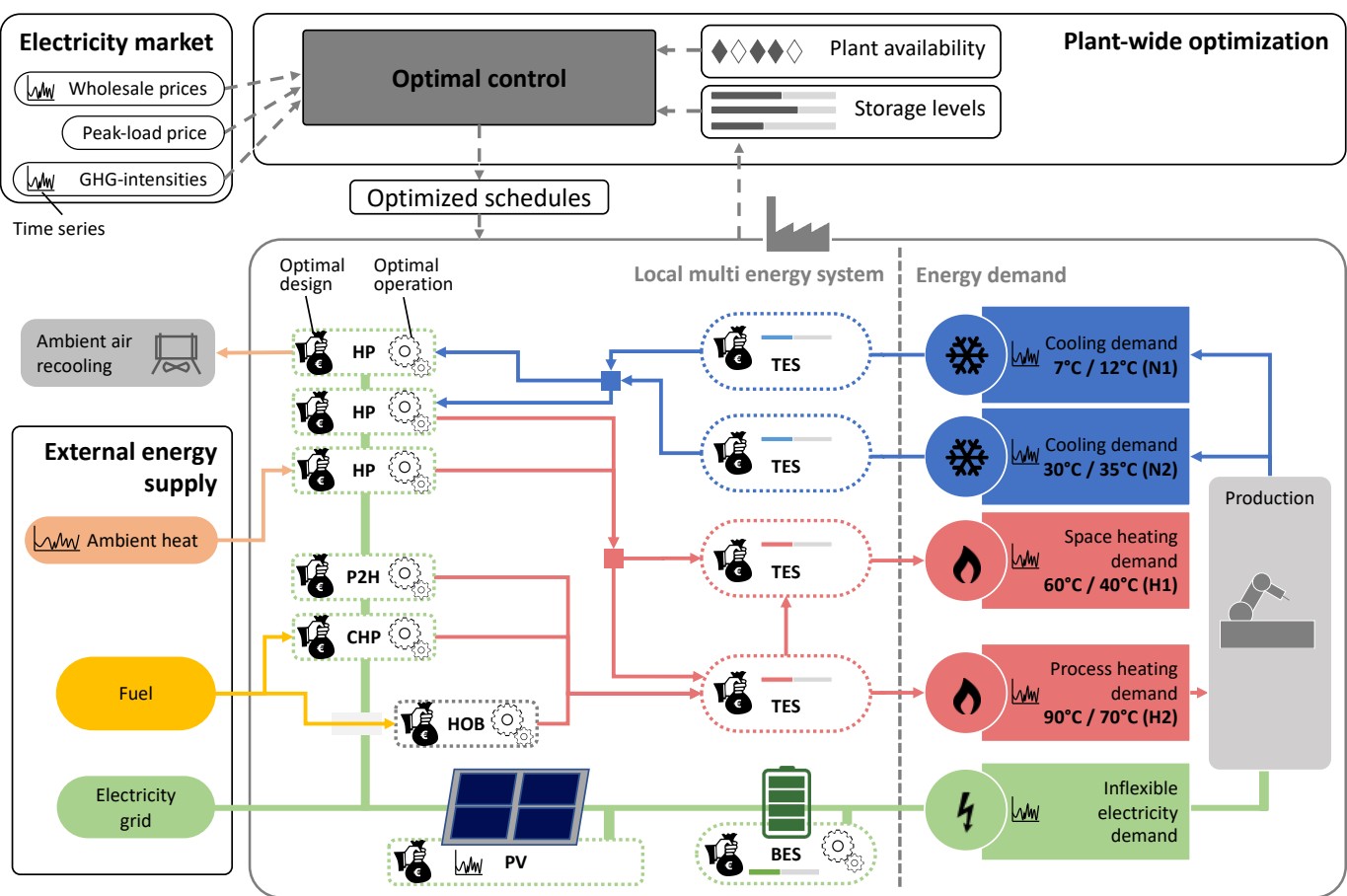

**Figure 12.** Setup and superstructure of Case Study 3.

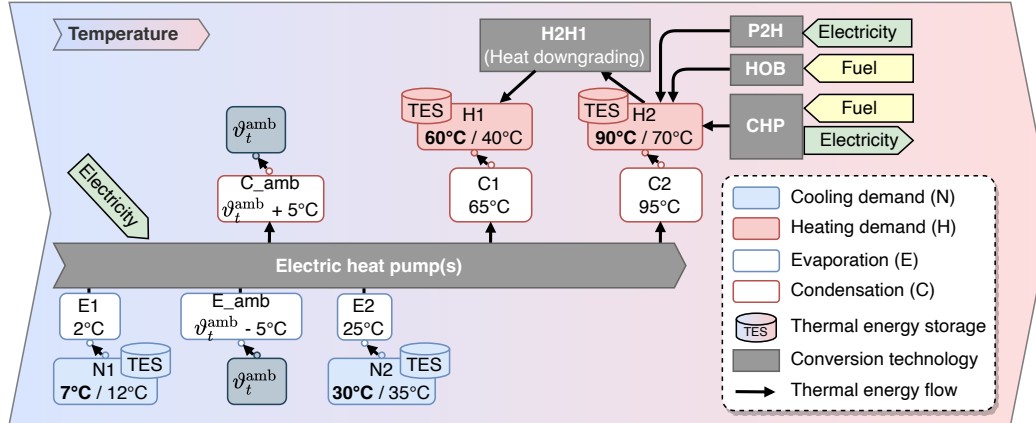

**Figure 13.** Scheme with details on modeling different temperature levels for Case Study 3. The HPs could choose between three source and sink temperature levels. The evaporation and condensation temperatures were calculated assuming a 5 K temperature difference for heat exchange. The coefficient of performance was calculated from the evaporation and condensation temperatures. Assuming the installation of multiple HPs, multiple parallel operation modes, i.e., temperature combinations between evaporation and condensation, can exist; however, for the calculation of the annualized investment costs, the capacities of all HPs are aggregated, which significantly reduces the model's complexity. For more details on HP modeling, see Appendix B.10.

We modeled seven scenarios: The reference scenario REF and the six scenarios sc2–sc7. REF only allows heat-only-boilers (HOBs) and cooling machines that are modeled using the HP component by allowing only heat transfers from the cooling demands to the ambient temperature.

Scenarios sc2 to sc7 allow all technologies of the superstructure and all HP operating modes. They differ from each other only by the Pareto weighting factor $\alpha$. The $\alpha$ values for the scenarios sc2 to sc7 were 0, 0.2, 0.4, 0.6, 0.8, and 1, respectively; i.e., sc2 optimized TAC ($\alpha = 0$), scenarios sc3 to sc6 optimized Pareto efficiency ($0 < \alpha < 1$), and sc7 optimized carbon emissions ($\alpha = 1$).

Figure 14 shows the TAC and annual carbon emissions of the resulting scenarios; and Figure 15 shows the resulting capacities, cost types, and distribution of the electricity exchange with the electricity grid. Due to the multi-objective optimization, scenarios sc3 to sc7 are Pareto-efficient—i.e., one objective value cannot be decreased without increasing the other. One can see that with increasing $\alpha$ values, i.e., increasing the focus on carbon emissions within the objective function, the investment cost and the annualized investment cost increase too. Compared to REF, sc2 and sc3 have lower TACs than REF by 36% and 9%, respectively, since higher annualized investment and maintenance costs are overcompensated by savings in operating costs. Additionally, sc2 and sc3 can reduce carbon emissions by 49% and 66%, respectively, compared to REF. Comparing sc2 and sc3 to REF produces no conflict of objectives, since REF is not on the Pareto frontier. The scenarios sc4 to sc7 have no economic advantage over REF; however, they do have an environmental advantage over REF. Scenarios sc3 and sc5 can be regarded as good trade-offs when looking at all available Pareto-efficient solutions. Scenario sc7 represents the highest possible carbon emission savings with 87%; however, the TACs are 41 times higher than in REF. Since in sc7 TACs are not considered, capacities were set to the highest possible values that are in place for each technology, e.g., 1 GWh for BES and 100 MWh for each TES. This is an unrealistic behavior that could be avoided by considering scope 3 emissions, which include the carbon emissions of the production of the energy technologies that are analogous to the annualized investment costs within the calculation of TACs. However, it demonstrates how scope 1 and scope 2 emissions can be reduced by increasing the system flexibility. As can be seen in Figure 15 bottom, in sc7 the technical upper limit of 20 MW of electrical power drawn from the grid is exploited during hours of low CEFs, which would stabilize the grid when

there is a surplus of renewable energy. Screenshots of Sankey diagrams for REF and sc3 are shown in Figure A4.

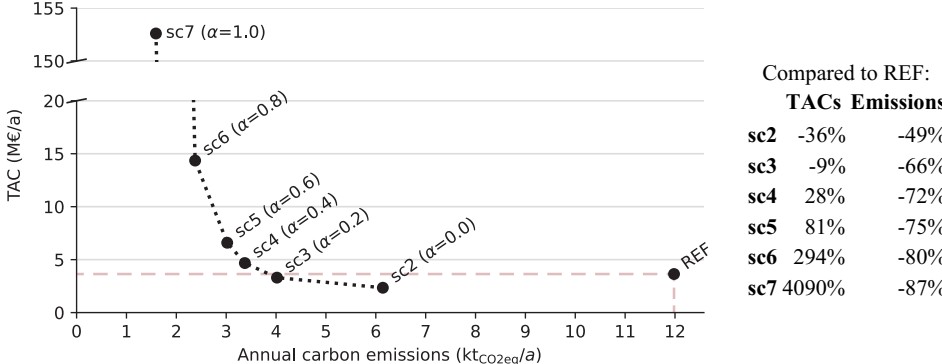

**Figure 14.** Pareto-optimal configuration scenarios of Case Study 3. The dotted line approximates the Pareto frontier. REF is not on the Pareto front, since both objectives can be improved as, e.g., in sc2 or sc3. The broken *y*-axis was used to fit in the minimal-emission scenario sc7 ($\alpha = 1$), which has a more than 41 times higher TAC than REF. Scenario sc3 has 9% less TAC than the REF whilst also reducing carbon emissions by two thirds.

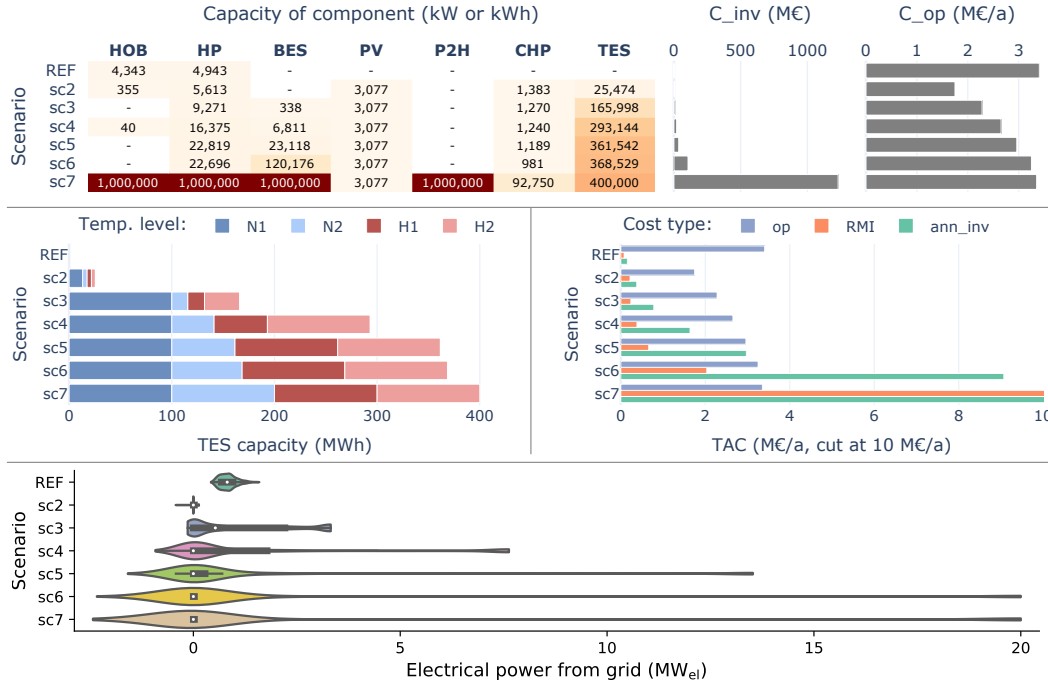

**Figure 15.** Results per scenario of Case Study 3. **Top**: Capacities, investment costs, and operating costs. **Middle left**: Thermal energy storage (TES) capacities per temperature level. **Middle right**: TAC per cost type: operating costs (op), maintenance costs (RMI), and annualized investment costs (ann_inv). **Bottom**: Distribution of the electricity bought from the grid (negative = sold electricity).

## 4. Conclusions

We developed, described, and demonstrated DRAF, an open-source multi-objective decision support tool for L-MESs. By providing vital information about the environmental and economical potential of innovative energy technologies and services, DRAF lowers investment barriers of L-MES decision-makers. It considers load flexibility, energy efficiency, electrification, and their interdependencies in an integrated model without neglecting critical aspects, such as multiple temperature levels or DR of production processes. DRAF considers flexible electricity sources and sinks across the whole energy conversion chain of an L-MES, such as an industrial site. While providing useful pre-configured components

that allow complex DR analyses with just a few lines of code, the tool setup does not restrict the users' freedom to build any possible MILP model. DRAF is needed since the existing software for potential identification of multi-energy systems is either (a) too generic to be practically applied to L-MESs, (b) leaves out essential aspects such as temperature levels or the access to dynamic emission factors, or (c) is not open-source.

Three case studies demonstrate how different settings and applications can be modeled within DRAF. Case Study 1 shows how price-based DR of a production process can reduce costs and operating carbon emissions. Case Study 2 demonstrates a simple design and operational optimization problem for a PV and multi-use BES system. The more sophisticated design optimization of Case Study 3 demonstrates the consideration of multiple temperature levels, the selection of heat sources and sinks for the HP, and the results of a multi-objective Pareto analysis to select the optimal trade-off between economic considerations and the reduction of carbon emissions.

This paper shows only a small sample of the possibilities of DRAF. Future work is, therefore, the application of the framework for a detailed analysis and optimization of a real L-MES, such as an industrial company. The implementation of myopic and stochastic modeling in a rolling horizon fashion, tools for scenario generation and reduction, and the selection of typical days are also future work.

**Author Contributions:** Conceptualization, M.F. and M.D.M.; methodology, M.F., M.B. (Markus Bohlayer) and M.D.M.; software, M.F.; investigation, M.F.; data curation, M.F.; writing—original draft preparation, M.F.; writing—review and editing, M.B. (Markus Bohlayer), M.B. (Marco Braun) and M.D.M.; visualization, M.F.; supervision, M.D.M.; project administration, M.B. (Marco Braun) and M.D.M.; funding acquisition, M.B. (Marco Braun) and M.D.M. All authors have read and agreed to the published version of the manuscript.

**Funding:** This research was performed as part of the MeSSO Research Group at the Munster Technological University (MTU) and in relation to the project WIN4climate as part of the National Climate Initiative financed by the Federal Ministry for Economic Affairs and Climate Action (BMWK) on the basis of a decision by the German Bundestag (number 03KF0094A). It was additionally funded by the MTU Risam scholarship scheme.

**Institutional Review Board Statement:** Not applicable.

**Informed Consent Statement:** Not applicable.

**Data Availability Statement:** Not applicable.

**Conflicts of Interest:** The authors declare no conflict of interest.

## Nomenclature

For a description of component-related symbols including units, please see description tables in Appendix B.

*Acronyms*

| | |
|---|---|
| CEF | Carbon emission factor |
| COP | Coefficient of performance |
| DR | Demand response |
| DRAF | Demand response analysis framework |
| HP | Electric heat pump |
| L-MES | Local multi-energy system |
| MEF | Marginal emission factor |
| MILP | Mixed-integer linear programming |
| PBDR | Price-based demand response |
| RES | Renewable energy sources |
| RTP | Real-time prices |
| TAC | Total annualized cost |
| TOU | Time of use |
| XEF | Grid mix emission factor |

*Component Labels*

| | |
|---|---|
| bes | Battery energy storage |
| bev | Battery electric vehicle |
| cdem | Cooling demand |
| chp | Combined heat and power |
| edem | Electricity demand |
| eg | Electricity grid |
| fuel | Fuels |
| h2h | Heat downgrading |
| hdem | Heat demand |
| hob | Heat-only boiler |
| hp | Electric heat pumps |
| p2h | Power-to-heat |
| pp | Production process |
| ps | Product storage |
| pv | Photovoltaic system |
| tes | Thermal energy storage |

*Symbols*

| | |
|---|---|
| $A$ | Area |
| $C$ | Costs |
| $c$ | Specific costs |
| CE | Carbon emissions |
| ce | Specific carbon emissions |
| cop | Coefficient of performance |
| $\dot{G}$ | Product flow |
| $\Delta_t$ | Time step |
| $\dot{Q}$ | Heat flow |
| $E$ | Electrical energy |
| $\eta$ | Efficiency |
| $F$ | Fuel flow |
| $G$ | Product |
| $k$ | A ratio |
| $n$ | A natural number |
| $N$ | Operation life |
| $P$ | Electrical power |
| $Q$ | Thermal energy |
| $T$ | Temperature |
| $y$ | Binary indicator |

*Superscripts*

| | |
|---|---|
| capn | New capacity |
| capx | Existing capacity |
| cond | Condensation |
| eva | Evaporation |
| fi | Feed-in |
| minpl | Minimal part load |
| oc | Own consumption |
| rmi | Repair, maintenance, and inspection |

*Indices and Sets*

| | |
|---|---|
| $c \in \mathcal{C}$ | Condensation temperature levels |
| $f \in \mathcal{F}$ | Fuel types |
| $h \in \mathcal{H}$ | Heating temperature levels |
| $i \in \mathcal{I}$ | Flow types |
| $j \in \mathcal{J}$ | Technology components |
| $l \in \mathcal{L}$ | Thermal demand temperature levels |
| $n \in \mathcal{N}$ | Cooling temperature levels |
| $t \in \mathcal{T}$ | Time steps |

## Appendix A. Screenshots of DRAF Output

*Appendix A.1. DemandAnalyzer*

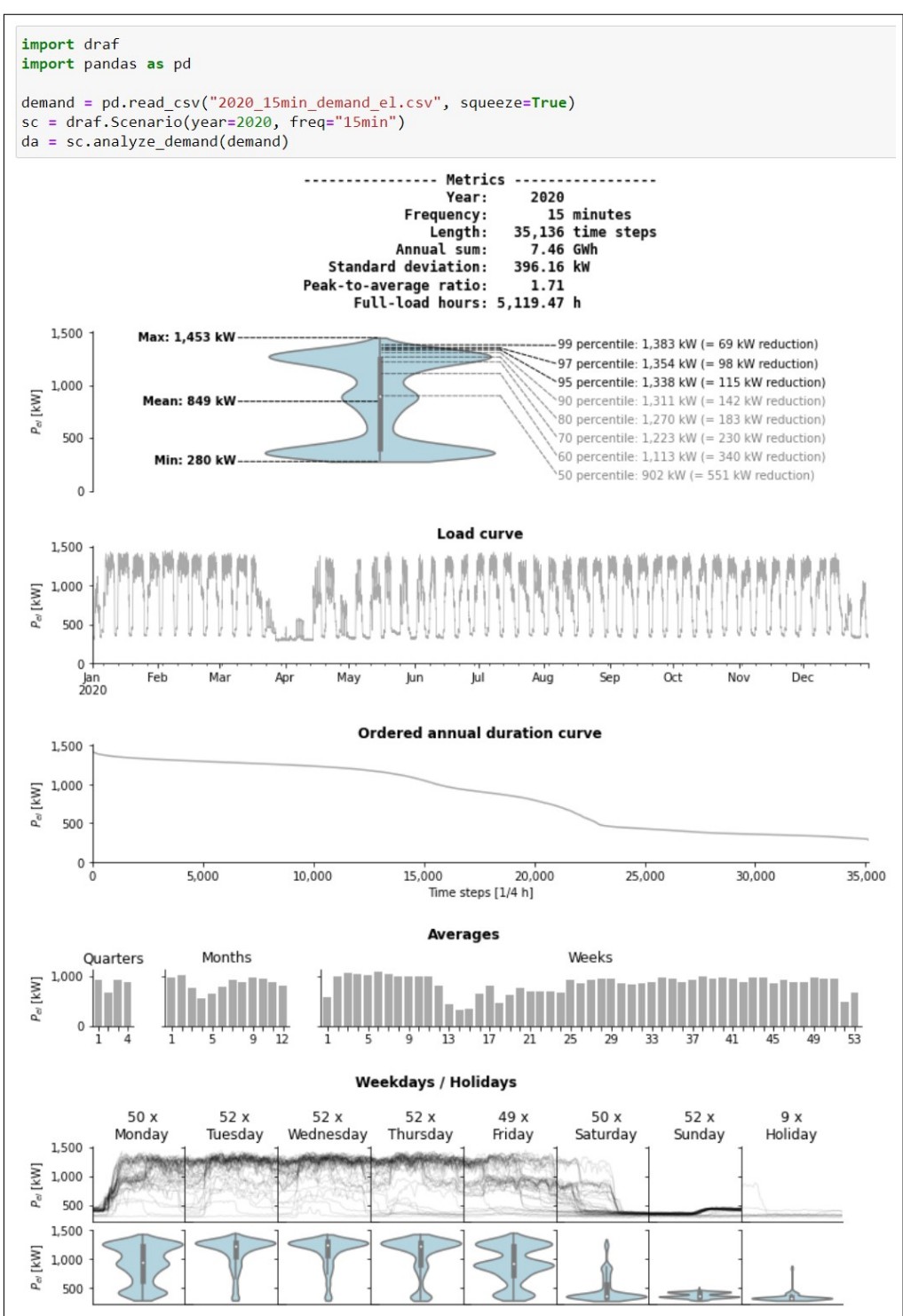

**Figure A1.** Demonstration of DemandAnalyzer.

*Appendix A.2. PeakLoadAnalyzer*

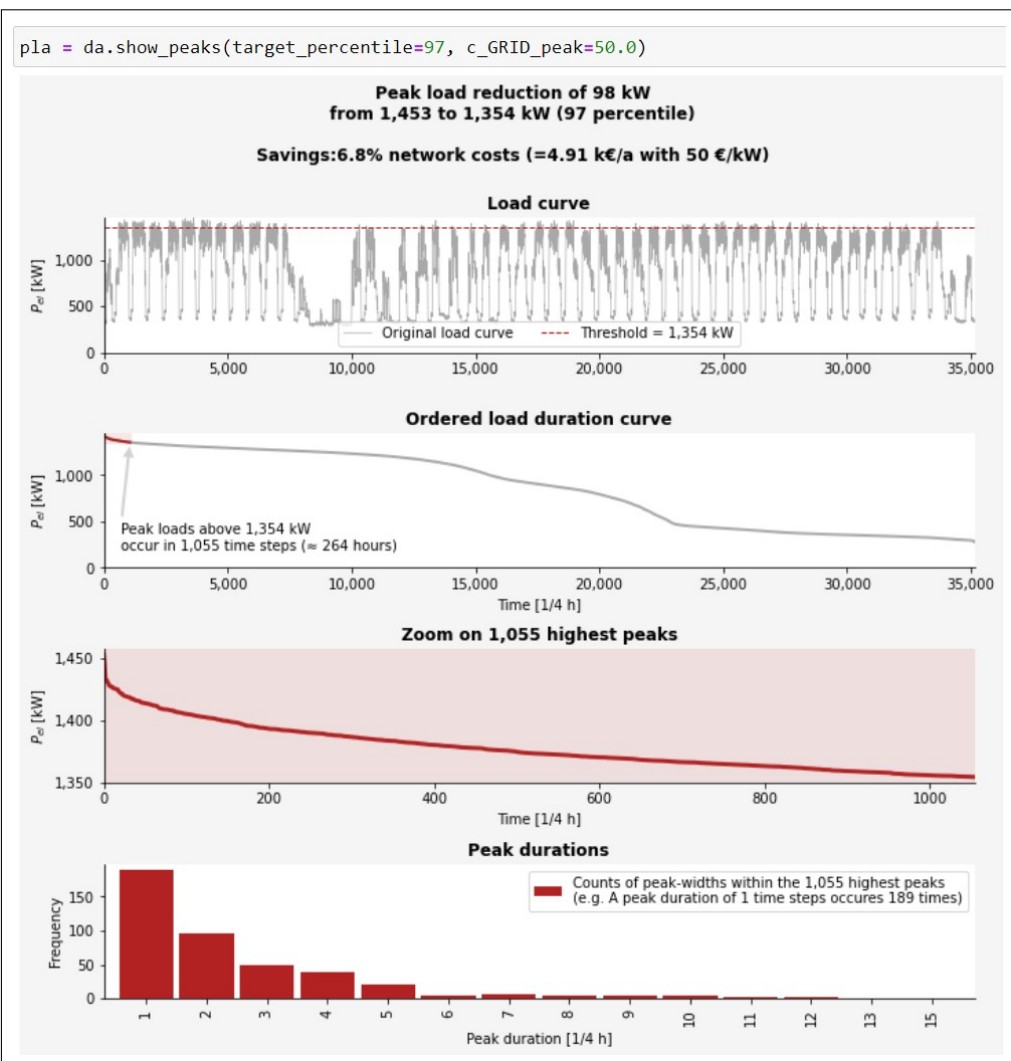

**Figure A2.** Demonstration of PeakLoadAnalyzer.

*Appendix A.3. Result Visualization*

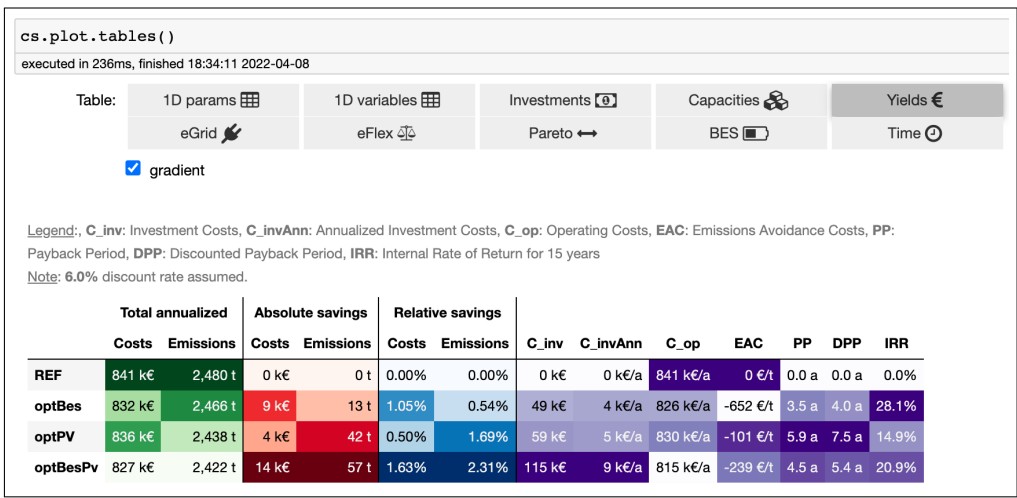

**Figure A3.** Screenshot of draf with overview of results of Case Study 2.

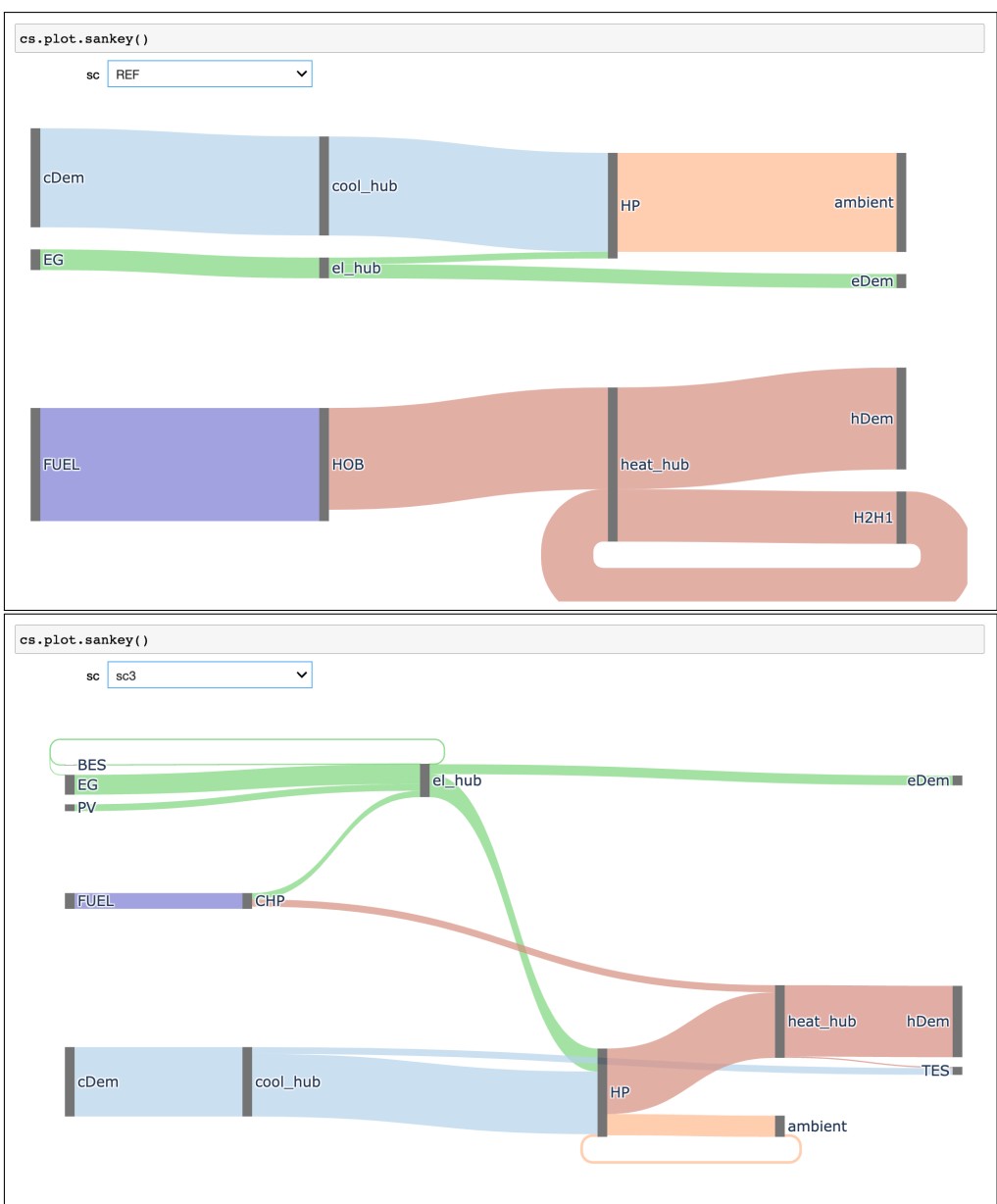

**Figure A4.** Screenshots of interactive Sankey diagrams. Data: Results of scenarios REF (top) and sc3 (bottom) of Case Study 3.

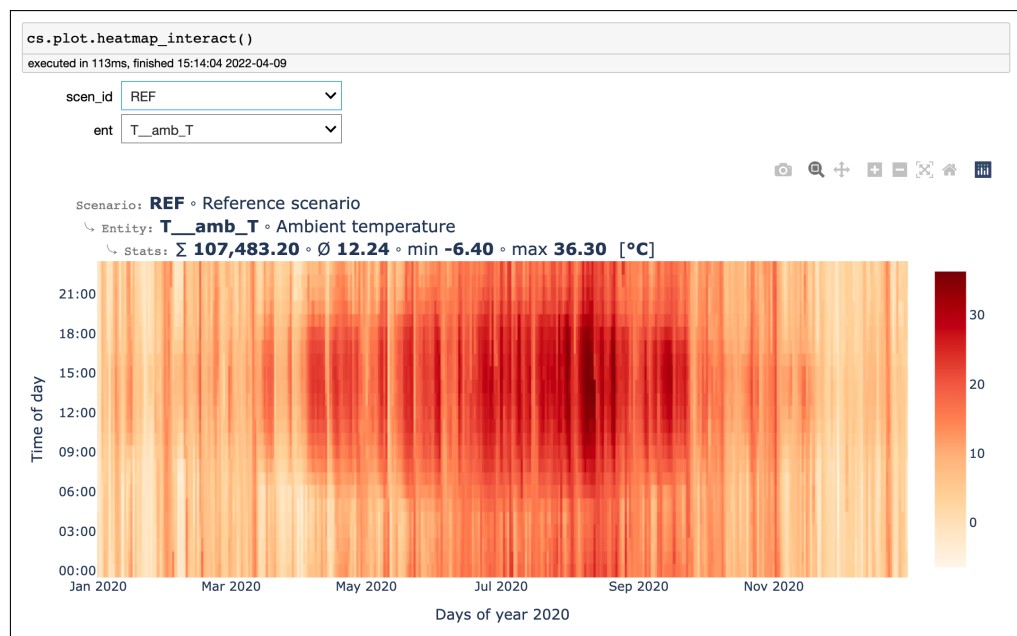

**Figure A5.** Screenshot of interactive heat map plotting of Case Study 3.

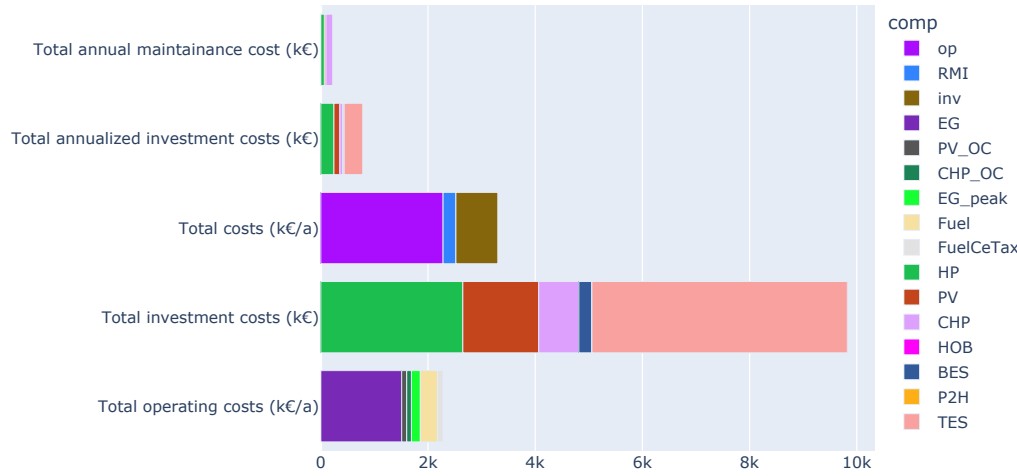

**Figure A6.** Output of `cs.scens.sc3.plot.collectors(filter_etype="C")` of Case Study 3.

## Appendix B. Component Templates Definitions

This section presents the mathematical formuation of component templates which can be classified as storages, conversion technologies, demands, interfaces, and a combination of them. For each technology, it contains an entity description table, the constraints, and the registrations to collectors. In the entity description table, all entities (parameters and variables) are listed with a source, unit, description, and default value(s) in the case of a scalar parameter. Thanks to consistent usage of naming conventions, the tables could programmatically be generated from the DRAF components, which also ensures consistency between the software and the paper. Furthermore, it demonstrates DRAF's possibilities in handling metadata such as units, data source information, and docstrings.

General parameters are: $\Delta_t$ is the width of the according time step in hours.

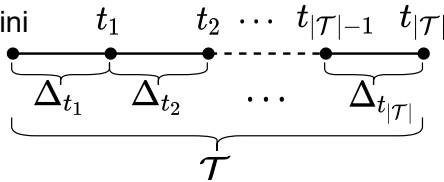

**Figure A7.** Relationship between time steps and time points.

*Appendix B.1. Electricity Grid (EG)*

| Symbol | Default | Src | Unit | Description |
|---|---|---|---|---|
| $ce_t^{eg}$ | - | | $kg_{CO2eq}/kWh_{el}$ | Carbon emission factors (via elmada using year, freq, country, and CEF-method) |
| $c^{eg,addon}$ | 0.131 | | $€/kWh_{el}$ | Electricity taxes and levies |
| $c^{eg,buypeak}$ | 50.000 | | $€/kW_{el}/a$ | Peak price |
| $c_t^{eg,flat}$ | - | | $€/kWh_{el}$ | Flat-electricity tariff (calculated from Real-time-price) |
| $c_t^{eg,rtp}$ | - | | $€/kWh_{el}$ | Day-ahead-market-prices (via elmada using year, freq, and country) |
| $c_t^{eg,tou}$ | - | | $€/kWh_{el}$ | Time-Of-Use-tariff (calculated from Real-time-price) |
| $c_t^{eg}$ | - | | $€/kWh_{el}$ | Chosen electricity tariff |
| $P^{eg,buypeak}$ | - | | $kW_{el}$ | Peak electrical power |
| $P_t^{eg,buy}$ | - | | $kW_{el}$ | Purchased electrical power |
| $P_t^{eg,sell}$ | - | | $kW_{el}$ | Selling electrical power |

$$P^{eg,buypeak} \geq P_t^{eg,buy} \tag{A1}$$

$$P_t^{eg,sell} = \sum_j \Phi_{i=elSell,j,t}^{source} \tag{A2}$$

$$\Phi_{i=el,j=eg,t}^{source} = P_t^{eg,buy}$$

$$\Phi_{i=el,j=eg,t}^{sink} = P_t^{eg,sell}$$

$$C_{j=eg}^{op} = 10^{-3} \sum_t \Delta_t \left( P_t^{eg,buy}(c_t^{eg} + c^{eg,addon}) - P_t^{eg,sell} c_t^{eg} \right)$$

$$C_{j=egPeak}^{op} = 10^{-3} P^{eg,buypeak} c^{eg,buypeak}$$

$$CE_{j=eg} = \sum_t \Delta_t (P_t^{eg,buy} - P_t^{eg,sell}) ce_t^{eg}$$

*Appendix B.2. Fuels (Fuel)*

| Symbol | Default | Src | Unit | Description |
|---|---|---|---|---|
| $F_f^{fuel}$ | - | | kW | Total fuel consumption |

$$F_f^{fuel} = \sum_j F_{f,j} \tag{A3}$$

$$C_{j=fuel}^{op} = 10^{-3} \sum_f F_f^{fuel} c_f^{fuel}$$

$$CE_{j=fuel} = \sum_f F_f^{fuel} ce_f^{fuel}$$

$$C^{\text{op}}_{j=\text{fuelTax}} = 10^{-3} CE_{j=\text{fuel}} c^{\text{fuelTax}}$$

*Appendix B.3. Battery Energy Storage (BES)*

| Symbol | Default | Src | Unit | Description |
|---|---|---|---|---|
| $E^{\text{bes,capx}}$ | 0.000 | | kWh$_{\text{el}}$ | Existing capacity |
| $N^{\text{bes}}$ | 20.000 | [84] | a | Operation life |
| $\eta^{\text{bes,ch}}$ | 97.468 | [85] | % | Charging efficiency |
| $\eta^{\text{bes,dis}}$ | 97.468 | [85] | % | Discharging efficiency |
| $\eta^{\text{bes,time}}$ | 99.998 | [86] | %/h | Efficiency due to self-discharge rate |
| $c^{\text{bes,inv}}$ | 720.000 | [87] | €/kWh$_{\text{el}}$ | CAPEX |
| $k^{\text{bes,ini}}$ | 0.000 | | % | Initial and final energy filling share |
| $k^{\text{bes,inpercap}}$ | 70.000 | [88] | % | Maximum charging power per capacity |
| $k^{\text{bes,outpercap}}$ | 70.000 | [88] | % | Maximum discharging power per capacity |
| $k^{\text{bes,rmi}}$ | 2.000 | [84] | % | Repair, maintenance, and inspection per year and investment cost |
| $E^{\text{bes,capn}}$ | - | | kWh$_{\text{el}}$ | New capacity |
| $E^{\text{bes}}_t$ | - | | kWh$_{\text{el}}$ | Electricity stored |
| $P^{\text{bes,in}}_t$ | - | | kW$_{\text{el}}$ | Charging power |
| $P^{\text{bes,out}}_t$ | - | | kW$_{\text{el}}$ | Discharging power |

$$E^{\text{bes}}_t = \eta^{\text{bes,time}} \begin{cases} k^{\text{bes,ini}}(E^{\text{bes,capx}} + E^{\text{bes,capn}}) & \text{if } t = t_0 \\ E^{\text{bes}}_{t-1} & \text{otherwise} \end{cases} \tag{A4}$$
$$d + \Delta_t \left( \eta^{\text{bes,ch}} P^{\text{bes,in}}_t - \frac{1}{\eta^{\text{bes,dis}}} P^{\text{bes,out}}_t \right)$$

$$E^{\text{bes}}_t \leq E^{\text{bes,capx}} + E^{\text{bes,capn}} \tag{A5}$$

$$P^{\text{bes,out}}_t \leq k^{\text{bes,outpercap}}(E^{\text{bes,capx}} + E^{\text{bes,capn}}) \tag{A6}$$

$$P^{\text{bes,in}}_t \leq k^{\text{bes,inpercap}}(E^{\text{bes,capx}} + E^{\text{bes,capn}}) \tag{A7}$$

$$E^{\text{bes}}_{t_{|\mathcal{T}|}} = k^{\text{bes,ini}}(E^{\text{bes,capx}} + E^{\text{bes,capn}}) \tag{A8}$$

$$\Phi^{\text{source}}_{i=\text{el},j=\text{bes},t} = P^{\text{bes,out}}_t$$

$$\Phi^{\text{sink}}_{i=\text{el},j=\text{bes},t} = P^{\text{bes,in}}_t$$

$$C^{\text{invAnn}}_{j=\text{bes}} = 10^{-3} E^{\text{bes,capn}} c^{\text{bes,inv}} k^{\text{af}}(r, N^{\text{bes}})$$

$$C^{\text{rmi}}_{j=\text{bes}} = 10^{-3} E^{\text{bes,capn}} c^{\text{bes,inv}} k^{\text{bes,rmi}}$$

Battery degradation is not considered. Please see [89] for more details regarding battery degradation in DR scenarios. A feed-in from the BES is not allowed. Following [90], $\eta^{\text{bes,ch}}$ and $\eta^{\text{bes,dis}}$ are calculated as the square root of the cycle efficiency, assuming them to be symmetrical.

*Appendix B.4. Thermal Energy Storage (TES)*

| Symbol | Default | Src | Unit | Description |
|---|---|---|---|---|
| $N^{\text{tes}}$ | 30.000 | [78] | a | Operation life |
| $Q_l^{\text{tes,capx}}$ | - | | $\text{kWh}_{\text{th}}$ | Existing capacity |
| $\eta^{\text{tes,time}}$ | 99.500 | | % | Storing efficiency |
| $c^{\text{tes,inv}}$ | 28.709 | [91] | $\text{€/kW}_{\text{th}}$ | CAPEX |
| $k_l^{\text{tes,ini}}$ | - | | % | Initial and final energy level share |
| $k^{\text{tes,inpercap}}$ | 50.000 | | % | Ratio loading power/capacity |
| $k^{\text{tes,outpercap}}$ | 50.000 | | % | Ratio loading power/capacity |
| $k^{\text{tes,rmi}}$ | 0.100 | [91] | % | Repair, maintenance, and inspection per year and investment cost |
| $Q_l^{\text{tes,capn}}$ | - | | $\text{kWh}_{\text{th}}$ | New capacity |
| $Q_{t,l}^{\text{tes}}$ | - | | $\text{kWh}_{\text{th}}$ | Stored heat |
| $\dot{Q}_{t,l}^{\text{tes,in}}$ | - | | $\text{kW}_{\text{th}}$ | Storage input heat flow |

$$Q_{t,l}^{\text{tes}} = \eta^{\text{tes,time}} \begin{cases} k_l^{\text{tes,ini}}(Q_l^{\text{tes,capx}} + Q_l^{\text{tes,capn}}) & \text{if } t = t_0 \\ Q_{t-1,l}^{\text{tes}} & \text{otherwise} \end{cases} \\ + \Delta_t(\eta^{\text{tes,cycle}} \dot{Q}_{t,l}^{\text{tes,in}} - \dot{Q}_{t,l}^{\text{tes,out}}) \tag{A9}$$

$$Q_{t,l}^{\text{tes}} \leq Q_l^{\text{tes,capx}} + Q_l^{\text{tes,capn}} \tag{A10}$$

$$\dot{Q}_{t,l}^{\text{tes,in}} \leq k^{\text{tes,inpercap}}(Q_l^{\text{tes,capx}} + Q_l^{\text{tes,capn}}) \tag{A11}$$

$$\dot{Q}_{t,l}^{\text{tes,out}} \leq k^{\text{tes,outpercap}}(Q_l^{\text{tes,capx}} + Q_l^{\text{tes,capn}}) \tag{A12}$$

$$Q_{t,l}^{\text{tes}} = k_l^{\text{tes,ini}}(Q_l^{\text{tes,capx}} + Q_l^{\text{tes,capn}}) \quad \forall t \in \{t_1, t_{|\mathcal{T}|}\} \tag{A13}$$

$$\dot{Q}_{t,l}^{\text{tes,in}} \in \mathbb{R}^{|T| \times |L|}$$

$$\Phi_{i,j=\text{tes},t}^{\text{source}} = \dot{Q}_{t,l}^{\text{tes,out}} \quad \forall i, l \in \mathcal{L}$$

$$\Phi_{i,j=\text{tes},t}^{\text{sink}} = \dot{Q}_{t,l}^{\text{tes,in}} \quad \forall i, l \in \mathcal{L}$$

$$C_{j=\text{tes}}^{\text{invAnn}} = 10^{-3} \sum_l Q_l^{\text{tes,capn}} c^{\text{tes,inv}} k^{\text{af}}(r, N^{\text{tes}})$$

$$C_{j=\text{tes}}^{\text{rmi}} = 10^{-3} \sum_l Q_l^{\text{tes,capn}} c^{\text{tes,inv}} k^{\text{tes,rmi}}$$

*Appendix B.5. Photovoltaic System (PV)*

| Symbol | Default | Src | Unit | Description |
|---|---|---|---|---|
| $A^{\text{pv,avail}}$ | 100.000 | | $\text{m}^2$ | Area available for new PV |
| $A^{\text{pv,perpeak}}$ | 6.500 | | $\text{m}^2/\text{kW}_{\text{peak}}$ | Area efficiency of new PV |
| $N^{\text{pv}}$ | 25.000 | [92] | a | Operation life |
| $P^{\text{pv,capx}}$ | 0.000 | | $\text{kW}_{\text{peak}}$ | Existing capacity |
| $P_t^{\text{pv,profile}}$ | - | [72] | $\text{kW}_{\text{el}}/\text{kW}_{\text{peak}}$ | Produced PV-power for 1 $\text{kW}_{\text{peak}}$ |
| $c^{\text{pv,inv}}$ | 460.000 | [83] | $\text{€/kW}_{\text{peak}}$ | CAPEX |
| $c^{\text{pv,oc}}$ | 0.028 | [93] | $\text{€/kWh}_{\text{el}}$ | Renewable Energy Law (EEG) levy on own consumption |
| $k^{\text{pv,rmi}}$ | 2.000 | [92] | % | Repair, maintenance, and inspection per year and investment cost |
| $P^{\text{pv,capn}}$ | - | | $\text{kW}_{\text{peak}}$ | New capacity |
| $P_t^{\text{pv,fi}}$ | - | | $\text{kW}_{\text{el}}$ | Feed-in |
| $P_t^{\text{pv,oc}}$ | - | | $\text{kW}_{\text{el}}$ | Own consumption |

$$(P^{\text{pv,capx}} + \mathbf{P}^{\text{pv,capn}})P_t^{\text{pv,profile}} = \mathbf{P}_t^{\text{pv,fi}} + \mathbf{P}_t^{\text{pv,oc}} \tag{A14}$$

$$\mathbf{P}^{\text{pv,capn}} \leq \frac{A^{\text{pv,avail}}}{A^{\text{pv,perpeak}}} \tag{A15}$$

$$\boldsymbol{\Phi}_{i=\text{el},j,t}^{\text{source}} = \mathbf{P}_t^{\text{pv,oc}}$$

$$\mathbf{P}_{t,j=\text{pv},t}^{\text{sell}} = \mathbf{P}_t^{\text{pv,fi}}$$

$$C_{j=\text{pv}}^{\text{op}} = 10^{-3} c^{\text{pv,oc}} \sum_t \Delta_t \mathbf{P}_t^{\text{pv,oc}}$$

$$C_{j=\text{pv}}^{\text{invAnn}} = 10^{-3} \mathbf{P}^{\text{pv,capn}} c^{\text{pv,inv}} k^{\text{af}}(r, N^{\text{pv}})$$

$$C_{j=\text{pv}}^{\text{rmi}} = 10^{-3} \mathbf{P}^{\text{pv,capn}} c^{\text{pv,inv}} k^{\text{pv,rmi}}$$

### Appendix B.6. Battery Electric Vehicle (BEV)

| Symbol | Default | Src | Unit | Description |
|---|---|---|---|---|
| $E_b^{\text{bev,cap1bat}}$ | - | | kWh$_{\text{el}}$ | Capacity of one battery |
| $E_b^{\text{bev,capx}}$ | - | | kWh$_{\text{el}}$ | Capacity of all batteries |
| $P_{t,b}^{\text{bev,drive}}$ | - | | kW$_{\text{el}}$ | Power use |
| $\eta^{\text{bev,ch}}$ | 97.468 | [85] | % | Charging efficiency |
| $\eta^{\text{bev,dis}}$ | 97.468 | [85] | % | Discharging efficiency |
| $\eta^{\text{bev,time}}$ | 100.000 | | % | Storing efficiency. Must be 1.0 for the uncontrolled charging in REF |
| $k_b^{\text{bev,empty}}$ | - | | % | Minimum state of charge |
| $k_b^{\text{bev,full}}$ | - | | % | Maximum state of charge |
| $k_b^{\text{bev,ini}}$ | - | | % | Initial and final state of charge |
| $k_b^{\text{bev,inpercap}}$ | - | [88] | % | Maximum charging power per capacity |
| $k_b^{\text{bev,v2xpercap}}$ | - | [88] | % | Maximum v2x discharging power per capacity |
| $n_b^{\text{bev,nbats}}$ | - | | - | Number of batteries |
| $y_{t,b}^{\text{bev,avail}}$ | - | | - | If BEV is available for charging at time step |
| $z^{\text{bev,smart}}$ | 0.000 | | - | If smart charging is allowed |
| $z^{\text{bev,v2x}}$ | 0.000 | | - | If vehicle-to-X is allowed |
| $\mathbf{E}_{t,b}^{\text{bev}}$ | - | | kWh$_{\text{el}}$ | Electricity stored in BEV battery |
| $\mathbf{P}_{t,b}^{\text{bev,in}}$ | - | | kW$_{\text{el}}$ | Charging power |
| $\mathbf{P}_{t,b}^{\text{bev,v2x}}$ | - | | kW$_{\text{el}}$ | Discharging power for vehicle-to-X |
| $\mathbf{X}^{\text{bev,penalty}}$ | - | | - | Penalty to ensure uncontrolled charging in REF |

$$\mathbf{E}_{t,b}^{\text{bev}} = \eta^{\text{bev,time}} \begin{cases} k_b^{\text{bev,ini}} E_b^{\text{bev,capx}} & \text{if } t = t_0 \\ \mathbf{E}_{t-1,b}^{\text{bev}} & \text{otherwise} \end{cases} \\ + \Delta_t \left( \eta^{\text{bev,ch}} \mathbf{P}_{t,b}^{\text{bev,in}} - \frac{1}{\eta^{\text{bev,dis}}} (\mathbf{P}_{t,b}^{\text{bev,drive}} + \mathbf{P}_{t,b}^{\text{bev,v2x}}) \right) \tag{A16}$$

$$\mathbf{E}_{t,b}^{\text{bev}} \leq k_b^{\text{bev,full}} E_b^{\text{bev,capx}} \tag{A17}$$

$$\mathbf{E}_{t,b}^{\text{bev}} \geq k_b^{\text{bev,empty}} E_b^{\text{bev,capx}} \tag{A18}$$

$$E_b^{\text{bev,capx}} = n_b^{\text{bev,nbats}} E_b^{\text{bev,cap1bat}} \tag{A19}$$

$$\mathbf{P}_{t,b}^{\text{bev,in}} \leq y_{t,b}^{\text{bev,avail}} k_b^{\text{bev,inpercap}} E_b^{\text{bev,capx}} \tag{A20}$$

$$\mathbf{P}_{t,b}^{\text{bev,v2x}} \leq z^{\text{bev,v2x}} y_{t,b}^{\text{bev,avail}} k_b^{\text{bev,v2xpercap}} E_b^{\text{bev,capx}} \tag{A21}$$

$$\mathbf{E}_{t_{|\mathcal{T}|},b}^{\text{bev}} = k_b^{\text{bev,ini}} E_b^{\text{bev,capx}} \tag{A22}$$

$$X^{\text{bev,penalty}} = (1 - z^{\text{bev,smart}}) \sum_{t,b} t P_{t,b}^{\text{bev,in}} \tag{A23}$$

$$X_{j=\text{bev}}^{\text{penalty}} = X^{\text{bev,penalty}}$$

$$\Phi_{i=\text{el},j=\text{bev},t}^{\text{source}} = \sum_b P_{t,b}^{\text{bev,v2x}}$$

$$\Phi_{i=\text{el},j=\text{bev},t}^{\text{sink}} = \sum_b P_{t,b}^{\text{bev,in}}$$

*Appendix B.7. Combined Heat and Power (CHP)*

| Symbol | Default | Src | Unit | Description |
|---|---|---|---|---|
| $N^{\text{chp}}$ | 25.000 | [94] | a | Operation life |
| $P^{\text{chp,capx}}$ | 0.000 | | $\text{kW}_{\text{el}}$ | Existing capacity |
| $P^{\text{chp,max}}$ | 100,000.000 | | $\text{kW}_{\text{el}}$ | Big-M number (upper bound for CAPn + CAPx) |
| $\eta^{\text{chp,el}}$ | 40.000 | [95] | % | Electric efficiency |
| $\eta^{\text{chp,th}}$ | 45.000 | [95] | % | Thermal efficiency |
| $c^{\text{chp,inv}}$ | 589.458 | [96] | €/$\text{kW}_{\text{el}}$ | CAPEX |
| $c^{\text{chp,oc}}$ | 0.028 | [93] | €/$\text{kWh}_{\text{el}}$ | Renewable Energy Law (EEG) levy on own consumption |
| $k^{\text{chp,minpl}}$ | 50.000 | | % | Minimal allowed part load |
| $k^{\text{chp,rmi}}$ | 18.000 | [94] | % | Repair, maintenance, and inspection per year and investment cost |
| $F_{t,f}^{\text{chp}}$ | - | | kW | Consumed fuel flow |
| $P^{\text{chp,capn}}$ | - | | $\text{kW}_{\text{el}}$ | New capacity |
| $P_t^{\text{chp,fi}}$ | - | | $\text{kW}_{\text{el}}$ | Feed-in |
| $P_t^{\text{chp,oc}}$ | - | | $\text{kW}_{\text{el}}$ | Own consumption |
| $P_t^{\text{chp}}$ | - | | $\text{kW}_{\text{el}}$ | Producing power |
| $Y_t^{\text{chp}}$ | - | | - | Binary: If in operation |
| $\dot{Q}_t^{\text{chp}}$ | - | | $\text{kW}_{\text{th}}$ | Producing heat flow |

$$P_t^{\text{chp}} = \eta^{\text{chp,el}} \sum_f F_{t,f}^{\text{chp}} \tag{A24}$$

$$\dot{Q}_t^{\text{chp}} = \eta^{\text{chp,th}} \sum_f F_{t,f}^{\text{chp}} \tag{A25}$$

$$P_t^{\text{chp}} \leq P^{\text{chp,capx}} + P^{\text{chp,capn}} \tag{A26}$$

$$P_t^{\text{chp}} = P_t^{\text{chp,fi}} + P_t^{\text{chp,oc}} \tag{A27}$$

$$P_t^{\text{chp}} \leq Y_t^{\text{chp}} P^{\text{chp,max}} \tag{A28}$$

$$P_t^{\text{chp}} \geq k^{\text{chp,minpl}} (P^{\text{chp,capx}} + P^{\text{chp,capn}}) - P^{\text{chp,max}} (1 - Y_t^{\text{chp}}) \tag{A29}$$

$$Y_t^{\text{chp}} \in \{0, 1\}$$

$$\Phi_{i=\text{heat2},j=\text{chp},t}^{\text{source}} = \dot{Q}_t^{\text{chp}}$$

$$\Phi_{i=\text{el},j=\text{chp},t}^{\text{source}} = P_t^{\text{chp,oc}}$$

$$P_{t,j=\text{chp}}^{\text{sell}} = P_t^{\text{chp,fi}}$$

$$F_{f,j=\text{chp}} = \sum_t \Delta_t F_{t,f}^{\text{chp}}$$

$$C_{j=\text{chp}}^{\text{op}} = 10^{-3} c^{\text{chp,oc}} \sum_t \Delta_t P_t^{\text{chp,oc}}$$

$$C_{j=\text{chp}}^{\text{invAnn}} = 10^{-3} P^{\text{chp,capn}} c^{\text{chp,inv}} k^{\text{af}}(r, N^{\text{chp}})$$

$$C_{j=\text{chp}}^{\text{rmi}} = 10^{-3} P^{\text{chp,capn}} c^{\text{chp,inv}} k^{\text{chp,rmi}}$$

*Appendix B.8. Heat-Only Boiler (HOB)*

| Symbol | Default | Src | Unit | Description |
|---|---|---|---|---|
| $N^{\text{hob}}$ | 15.000 | [94] | a | Operation life |
| $\dot{Q}^{\text{hob,capx}}$ | 0.000 | | $kW_{\text{th}}$ | Existing capacity |
| $\eta^{\text{hob}}$ | 90.000 | [94] | % | Thermal efficiency |
| $c^{\text{hob,inv}}$ | 57.133 | [97] | €/$kW_{\text{th}}$ | CAPEX |
| $k^{\text{hob,rmi}}$ | 18.000 | [94] | % | Repair, maintenance, and inspection per year and investment cost |
| $F_{t,f}^{\text{hob}}$ | - | | kW | Input fuel flow |
| $\dot{Q}^{\text{hob,capn}}$ | - | | $kW_{\text{th}}$ | New capacity |
| $\dot{Q}_t^{\text{hob}}$ | - | | $kW_{\text{th}}$ | Ouput heat flow |

$$\dot{Q}_t^{\text{hob}} = \eta^{\text{hob}} \sum_f F_{t,f}^{\text{hob}} \tag{A30}$$

$$\dot{Q}_t^{\text{hob}} \leq \dot{Q}^{\text{hob,capx}} + \dot{Q}^{\text{hob,capn}} \tag{A31}$$

$$\Phi_{i=\text{heat2},j=\text{hob},t}^{\text{source}} = \dot{Q}_t^{\text{hob}}$$

$$F_{f,j=\text{hob}} = \sum_t \Delta_t F_{t,f}^{\text{hob}}$$

$$C_{j=\text{hob}}^{\text{invAnn}} = 10^{-3} \dot{Q}^{\text{hob,capn}} c^{\text{hob,inv}} k^{\text{af}}(r, N^{\text{hob}})$$

$$C_{j=\text{hob}}^{\text{rmi}} = 10^{-3} \dot{Q}^{\text{hob,capn}} c^{\text{hob,inv}} k^{\text{hob,rmi}}$$

*Appendix B.9. Power-to-Heat (P2H)*

| Symbol | Default | Src | Unit | Description |
|---|---|---|---|---|
| $N^{\text{p2h}}$ | 30.000 | | a | Operation life |
| $\dot{Q}^{\text{p2h,capx}}$ | 0.000 | | $kW_{\text{th}}$ | Existing capacity |
| $\eta^{\text{p2h}}$ | 90.000 | [98] | % | Efficiency |
| $c^{\text{p2h,inv}}$ | 100.000 | [99] | €/$kW_{\text{th}}$ | System CAPEX |
| $k^{\text{p2h,rmi}}$ | 0.000 | | % | Repair, maintenance, and inspection per year and investment cost |
| $P_t^{\text{p2h}}$ | - | | $kW_{\text{el}}$ | Consuming power |
| $\dot{Q}^{\text{p2h,capn}}$ | - | | $kW_{\text{th}}$ | New capacity |
| $\dot{Q}_t^{\text{p2h}}$ | - | | $kW_{\text{th}}$ | Producing heat flow |

$$\dot{Q}_t^{\text{p2h}} = \eta^{\text{p2h}} P_t^{\text{p2h}} \tag{A32}$$

$$\dot{Q}_t^{\text{p2h}} \leq \dot{Q}^{\text{p2h,capx}} + \dot{Q}^{\text{p2h,capn}} \tag{A33}$$

$$\Phi_{i=\text{heat2},j=\text{p2h},t}^{\text{source}} = \dot{Q}_t^{\text{p2h}}$$

$$\Phi_{i=\text{el},j=\text{p2h},t}^{\text{sink}} = P_t^{\text{p2h}}$$

$$C_{j=\text{p2h}}^{\text{invAnn}} = 10^{-3} \dot{Q}^{\text{p2h,capn}} c^{\text{p2h,inv}} k^{\text{af}}(r, N^{\text{p2h}})$$

$$C_{j=\text{p2h}}^{\text{rmi}} = 10^{-3} \dot{Q}^{\text{p2h,capn}} c^{\text{p2h,inv}} k^{\text{p2h,rmi}}$$

*Appendix B.10. Electric Heat Pump (HP)*

| Symbol | Default | Src | Unit | Description |
|--------|---------|-----|------|-------------|
| $N^{\text{hp}}$ | 18.000 | [100] | a | Operation life |
| $\dot{Q}^{\text{hp,capx}}$ | 0.000 | | $\text{kW}_{\text{th}}$ | Existing heating capacity |
| $\dot{Q}^{\text{hp,max}}$ | 100,000.000 | | $\text{kW}_{\text{th}}$ | Big-M number (upper bound for CAPn + CAPx) |
| $\eta^{\text{hp}}$ | 50.000 | [101] | % | Ratio of reaching the ideal COP (exergy efficiency) |
| $\vartheta_c^{\text{hp,cond}}$ | - | | °C | Condensation side temperature |
| $\vartheta_e^{\text{hp,eva}}$ | - | | °C | Evaporation side temperature |
| $c^{\text{hp,inv}}$ | 285.788 | [102] | €/$\text{kW}_{\text{el}}$ | CAPEX |
| $k^{\text{hp,rmi}}$ | 2.500 | [100] | % | Repair, maintenance, and inspection per year and investment cost |
| $n^{\text{hp}}$ | 1.000 | | - | Maximum number of parallel operation modes |
| $P_{t,e,c}^{\text{hp}}$ | - | | $\text{kW}_{\text{el}}$ | Consuming power |
| $Y_{t,e,c}^{\text{hp}}$ | - | | - | Binary: If source and sink are connected at time-step |
| $\dot{Q}^{\text{hp,capn}}$ | - | | $\text{kW}_{\text{th}}$ | New heating capacity |
| $\dot{Q}_{t,e,c}^{\text{hp,cond}}$ | - | | $\text{kW}_{\text{th}}$ | Heat flow released on condensation side |
| $\dot{Q}_{t,e,c}^{\text{hp,eva}}$ | - | | $\text{kW}_{\text{th}}$ | Heat flow absorbed on evaporation side |

$$\dot{Q}_{t,c,n}^{\text{hp,cond}} = \text{cop}_{t,c,n}^{\text{hp}} P_{t,c,n}^{\text{hp}} \tag{A34}$$

$$\dot{Q}_{t,c,n}^{\text{hp,cond}} = \dot{Q}_{t,c,n}^{\text{hp,eva}} + P_{t,c,n}^{\text{hp}} \tag{A35}$$

$$\dot{Q}_{t,c,n}^{\text{hp}} \leq Y_{t,c,n}^{\text{hp}} \dot{Q}^{\text{hp,max}} \tag{A36}$$

$$\sum_{c,n} \dot{Q}_{t,c,n}^{\text{hp}} \leq \dot{Q}^{\text{hp,capx}} + \dot{Q}^{\text{hp,capn}} \tag{A37}$$

$$\sum_c \sum_n Y_{t,c,n}^{\text{hp}} \leq n^{\text{hp}} \tag{A38}$$

$$\text{cop}_{t,c,n}^{\text{hp}} = \begin{cases} 100 & \text{if } \vartheta_{t,c}^{\text{hp,cond}} \leq \vartheta_n^{\text{hp,eva}} \\ \eta^{\text{hp}} \text{cop}_{t,c,n}^{\text{hp,carnot}} & \text{otherwise} \end{cases} \tag{A39}$$

$$\text{cop}_{t,c,n}^{\text{hp,carnot}} = \frac{\vartheta_{t,c}^{\text{hp,cond}} + 273}{\vartheta_{t,c}^{\text{hp,cond}} - \vartheta_n^{\text{hp,eva}}} \tag{A40}$$

$$Y_{t,c,n}^{\text{hp}} \in \{0,1\}$$

$$\Phi_{i,j=\text{hp},t}^{\text{source}} = \dot{Q}_{t,c,n}^{\text{hp,cond}} \quad \forall i, c \in \mathcal{H}$$

$$\Phi_{i,j=\text{hp},t}^{\text{sink}} = \dot{Q}_{t,c,n}^{\text{hp,eva}} \quad \forall i, n \in \mathcal{N}$$

$$\Phi_{i=\text{el},j=\text{hp},t}^{\text{sink}} = P_{t,c,n}^{\text{hp}}$$

$$C_{j=\text{hp}}^{\text{invAnn}} = 10^{-3} \dot{Q}^{\text{hp,capn}} c^{\text{hp,inv}} k^{\text{af}}(r, N^{\text{hp}})$$

$$C_{j=\text{hp}}^{\text{rmi}} = 10^{-3} \dot{Q}^{\text{hp,capn}} c^{\text{hp,inv}} k^{\text{hp,rmi}}$$

*Appendix B.11. Heat Downgrading (H2H1)*

| Symbol | Default | Src | Unit | Description |
|---|---|---|---|---|
| $\dot{Q}_t^{\text{h2h1}}$ | - | | kW$_{\text{th}}$ | Heat down-grading |

$$\Phi_{i=\text{heat1},j=\text{h2h1},t}^{\text{source}} = \dot{Q}_t^{\text{h2h1}}$$

$$\Phi_{i=\text{heat2},j=\text{h2h1},t}^{\text{sink}} = \dot{Q}_t^{\text{h2h1}}$$

*Appendix B.12. Product Demand (pDem)*

Formulation partly based on [81].

| Symbol | Default | Src | Unit | Description |
|---|---|---|---|---|
| $\dot{G}_{t,s}^{\text{pdem}}$ | - | | t/h | Product demand |

$$\Phi_{i=\text{prod},j=\text{pdem},t}^{\text{sink}} = \dot{G}_{t,s}^{\text{pdem}}$$

*Appendix B.13. Production Process (PP)*

| Symbol | Default | Src | Unit | Description |
|---|---|---|---|---|
| $P_m^{\text{PP,capx}}$ | - | | kW$_{\text{el}}$ | |
| $\eta_{s,m}^{\text{PP}}$ | - | | % | Production efficiency |
| $c^{\text{PP,sc}}$ | 10.000 | | €/change | Costs per sort change |
| $c^{\text{PP,su}}$ | 10.000 | | €/SU | Costs per start up |
| $k_m^{\text{PP,minpl}}$ | - | | % | Minimum part load |
| $y_{t,m}^{\text{PP,avail}}$ | - | | - | If machine is available at time step |
| $y_{s,m}^{\text{PP,compat}}$ | - | | - | If machine and sort is compatible |
| $C^{\text{PP,sc}}$ | - | | k€ | Total cost of sort change |
| $C^{\text{PP,su}}$ | - | | k€ | Total cost of start up |
| $P_{t,s,m}^{\text{PP}}$ | - | | kW$_{\text{el}}$ | Nominal power consumption of machine |
| $Y_{t,s,m}^{\text{PP,op}}$ | - | | - | Binary: If machine is in operation |
| $Y_{t,s,m}^{\text{PP,sc}}$ | - | | - | Binary: If sort has just changed |
| $Y_{t,m}^{\text{PP,su}}$ | - | | - | Binary: If machine just started up |
| $\dot{G}_{t,s,m}^{\text{PP}}$ | - | | t/h | Production of machine |

$$\dot{G}_{t,s,m}^{\text{PP}} = \eta_{s,m}^{\text{PP}} y_{s,m}^{\text{PP,compat}} y_{t,m}^{\text{PP,avail}} \tag{A41}$$

$$P_{t,s,m}^{\text{PP}} \leq Y_{t,s,m}^{\text{PP,op}} P_m^{\text{PP,capx}} \tag{A42}$$

$$P_{t,s,m}^{\text{PP}} \geq Y_{t,s,m}^{\text{PP,op}} k_m^{\text{PP,minpl}} P_m^{\text{PP,capx}} \tag{A43}$$

$$C^{\text{PP,su}} = 10^{-3} \sum_{t,m} Y_{t,m}^{\text{PP,su}} c^{\text{PP,su}} \tag{A44}$$

$$Y_{t,m}^{\text{PP,su}} \geq \sum_s Y_{t,s,m}^{\text{PP,op}} - \sum_s Y_{t-1,s,m}^{\text{PP,op}} \tag{A45}$$

$$C^{\text{PP,sc}} = 10^{-3} \sum_{t,m} Y_{t,m}^{\text{PP,sc}} c^{\text{PP,sc}} \tag{A46}$$

$$Y_{t,s,m}^{\text{PP,sc}} \geq Y_{t,s,m}^{\text{PP,op}} - Y_{t-1,s,m}^{\text{PP,op}} \quad \forall t \in \mathcal{T} \setminus \{t_0\} \tag{A47}$$

$$\sum_s Y_{t,s,m}^{\text{PP,op}} \leq 1 \tag{A48}$$

$$\Phi_{i=\text{prod},j=\text{pp},t}^{\text{source}} = \sum_m \dot{G}_{t,s,m}^{\text{PP}}$$

$$P^{\text{sink}}_{i=\text{prod},j=\text{pp},t} = \sum_{s,m} P^{\text{pp}}_{t,s,m}$$

$$C^{\text{op}}_{j=\text{pp}} = C^{\text{pp,sc}} + C^{\text{pp,su}}$$

*Appendix B.14. Product Storage (PS)*

| Symbol | Default | Src | Unit | Description |
|--------|---------|-----|------|-------------|
| $G^{\text{ps,capx}}_s$ | - | | t | Existing storage capacity of product |
| $N^{\text{ps}}$ | 50.000 | | a | Operation life |
| $c^{\text{ps,inv}}$ | 1000.000 | | €/t | Investment cost |
| $k^{\text{ps,ini}}_s$ | - | | % | Initial storage filling level |
| $k^{\text{ps,min}}_s$ | - | | % | Share of minimal required storage filling level |
| $E^{\text{ps,delta}}$ | - | | kWh$_{\text{el}}$ | Energy equivalent |
| $G^{\text{ps,capn}}_s$ | - | | t | New capacity |
| $G^{\text{ps,delta}}_s$ | - | | t | Final time step deviation from init |
| $G^{\text{ps}}_{t,s}$ | - | | t | Storage filling level |

$$G^{\text{ps}}_{t,s} \leq (G^{\text{ps,capx}}_s + G^{\text{ps,capn}}_s) \tag{A49}$$

$$G^{\text{ps}}_{t,s} \geq k^{\text{ps,min}}_s (G^{\text{ps,capx}}_s + G^{\text{ps,capn}}_s) \tag{A50}$$

$$G^{\text{ps}}_{t_{|\mathcal{T}|},s} = k^{\text{ps,ini}}_s (G^{\text{ps,capx}}_s + G^{\text{ps,capn}}_s) - G^{\text{ps,delta}}_s \tag{A51}$$

$$E^{\text{ps,delta}} = \sum_s G^{\text{ps,delta}}_s \max_m \frac{1}{\eta^{\text{pp}}_{s,m}} \tag{A52}$$

$$\Phi^{\text{sink}}_{i=\text{prod},j=\text{ps},t} = \frac{1}{\Delta_t} \begin{cases} \left( k^{\text{ps,ini}}_s (G^{\text{ps,capx}}_s + G^{\text{ps,capn}}_s) - G^{\text{ps}}_{t,s} \right) & \text{if } t = t_0 \\ \left( G^{\text{ps}}_{t-1,s} - G^{\text{ps}}_{t,s} \right) & \text{otherwise} \end{cases}$$

$$C^{\text{op}}_{j=\text{ps}} = 3 \times 10^{-3} \left( \frac{\sum_t c^{\text{eg}}_t}{|\mathcal{T}|} + c^{\text{eg,addon}} \right) E^{\text{ps,delta}}$$

$$C^{\text{invAnn}}_{j=\text{ps}} = 10^{-3} \sum_s \dot{G}^{\text{ps,capn}}_s c^{\text{ps,inv}} k^{\text{af}}(r, N^{\text{ps}})$$

*Appendix B.15. Cooling Demand (cDem)*

| Symbol | Default | Src | Unit | Description |
|--------|---------|-----|------|-------------|
| $\dot{Q}^{\text{cdem}}_{t,n}$ | - | | kW$_{\text{th}}$ | Cooling demand |
| $\vartheta^{\text{cdem,in}}_n$ | - | | °C | Cooling inlet temperature |
| $\vartheta^{\text{cdem,out}}_n$ | - | | °C | Cooling outlet temperature |

$$\Phi^{\text{source}}_{i,j=\text{cdem},t} = \dot{Q}^{\text{cdem}}_{t,n} \quad \forall i, n \in \mathcal{N}$$

*Appendix B.16. Heating Demand (hDem)*

| Symbol | Default | Src | Unit | Description |
|--------|---------|-----|------|-------------|
| $\dot{Q}^{\text{hdem}}_{t,h}$ | - | | kW$_{\text{th}}$ | Heating demand |
| $\vartheta^{\text{hdem,in}}_h$ | - | | °C | Heating inlet temperature |
| $\vartheta^{\text{hdem,out}}_h$ | - | | °C | Heating outlet temperature |

$$\Phi^{\text{sink}}_{i,j=\text{hdem},t} = \dot{Q}^{\text{hdem}}_{t,h} \quad \forall i, h \in \mathcal{H}$$

*Appendix B.17. Electricity Demand (eDem)*

| Symbol | Default | Src | Unit | Description |
|--------|---------|-----|------|-------------|
| $P_t^{\text{edem}}$ | - | | kW$_{\text{el}}$ | Electricity demand from standard load profile G3: Business continuous |

$$\Phi_{i=\text{el},j=\text{el},t}^{\text{sink}} = P_t^{\text{edem}}$$

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
