# Peer review of "Demand Response Analysis Framework (DRAF): An Open-Source Multi-Objective Decision Support Tool for Decarbonizing Local Multi-Energy Systems"

_sustainability, doi:10.3390/su14138025_

Round 1

Reviewer 1 Report

Paper no.: sustainability-1743951

Title: Demand Response Analysis Framework (DRAF): An open-source multi-objective decision support tool for decarbonizing local multi-energy systems

In the paper the Demand Response Analysis Framework as a new open-source Python decision support tool which allows to integrally optimizes the design and operation of energy technologies considering demand-side flexibility, electrification, and renewable energy source was proposed. It is well written paper worth for publication in Sustainability. Anyway some minor revision is required before acceptance of the paper:

·       In the introduction in the part where the description of the need of decarbonization is presented except the Paris Agreement (2015) and the Glasgow Climate Pact (2021) also the EU Green Deal should be also shortly described.

·        

·       What do you mean that the energy transition significantly depends on the investment and operational decisions of individual L-MES decision-makers?

·       Some information about the cost of implementation RES should be added in the revised version of the manuscript.

·       There are numerous paper dealing with the decarbonizing local multi-energy systems. Some short reviews of the existing study presented in the existing literature should be added in the revised version of the manuscript.

·       In the mathematical optimization models different complexity reduction methods can be applied to energy systems optimization to formulate MILP models. Why did you chose MILP assuming?

·       Figure 1 which present the external resources and core elements of DRAF and its link to the energy system analysis process in unclear and should be widely described in the revised version of the manuscript.

·       The information about the way how the paper is not needed and should be remove from the paper.

·       Functions with a simple PV example component should be presented in the Appendix.

·       In the conclusions please not repeat the information which were presented in the abstract and in the introduction part. The Conclusions should presented only the main points which were obtained in the analysis.

Please be more selective in citation. This is typical research paper but the references look like in the review paper. Moreover please avoid lump citations.

Reviewer 2 Report

In this paper, the authors proposed the Demand Response Analysis Framework (DRAF), which is a Python decision support framework. By considering demand-side flexibility, electrification, and renewable energy sources, the proposed tool can optimize the design and operation of energy technologies. This is well written and organized paper. It is scientifically sound and contains sufficient interest. The authors have made a good attempt, I think. This is a good paper. My comments are as follows:

  1. The authors should not use acronym without explanation. All acronyms must be defined before use. e.g. COMANDO (component-oriented modeling and optimization for nonlinear design and operation), NEMO (Next Energy Modeling system for Optimization), etc.
  2. In section 2, the explanation about the code and algorithm is not enough. The meaning of variables and functions is not clear. Readers will be confused. The authors should explain your idea mathematically.
  3. There is no left-hand in Eq. (1).

Round 2

Reviewer 1 Report

The paper was corrected according my recommendations.